# L1CAM is required for early dissemination of fallopian tube carcinoma precursors to the ovary

Kai Doberstein[1,8], Rebecca Spivak[1], Hunter D. Reavis[1], Jagmohan Hooda [1,9], Yi Feng[1], Paul T. Kroeger Jr[1], Sarah Stuckelberger[1], Gordon B. Mills [2], Kyle M. Devins[3], Lauren E. Schwartz[3], Marcin P. Iwanicki [4], Mina Fogel[5], Peter Altevogt[6] & Ronny Drapkin [1,7✉]

Most ovarian high-grade serous carcinomas (HGSC) arise from Serous Tubal Intraepithelial Carcinoma (STIC) lesions in the distal end of the fallopian tube (FT). Formation of STIC lesions from FT secretory cells leads to seeding of the ovarian surface, with rapid tumor dissemination to other abdominal structures thereafter. It remains unclear how nascent malignant cells leave the FT to colonize the ovary. This report provides evidence that the L1 cell adhesion molecule (L1CAM) contributes to the ability of transformed FT secretory cells (FTSEC) to detach from the tube, survive under anchorage-independent conditions, and seed the ovarian surface. L1CAM was highly expressed on the apical cells of STIC lesions and contributed to ovarian colonization by upregulating integrins and fibronectin in malignant cells and activating the AKT and ERK pathways. These changes increased cell survival under ultra-low attachment conditions that mimic transit from the FT to the ovary. To study dissemination to the ovary, we developed a tumor-ovary co-culture model. We showed that L1CAM expression was important for FT cells to invade the ovary as a cohesive group. Our results indicate that in the early stages of HGSC development, transformed FTSECs disseminate from the FT to the ovary in a L1CAM-dependent manner.

[1] Ovarian Cancer Research Center, University of Pennsylvania, Perelman School of Medicine, Philadelphia, PA 19104, USA. [2] Knight Cancer Institute, Oregon Health and Science University, Portland, OR 97239, USA. [3] Department of Pathology and Laboratory Medicine, University of Pennsylvania, Perelman School of Medicine, Philadelphia, PA 19104, USA. [4] Department of Bioengineering, Chemistry, Chemical Biology and Biological Sciences, Stevens Institute of Technology, Hoboken, NJ, USA. [5] Central Laboratories, Kaplan Medical Center, Rehovot, Israel. [6] Skin Cancer Unit, German Cancer Research Center (DKFZ), Heidelberg, Germany. [7] Basser Center for BRCA, Abramson Cancer Center, University of Pennsylvania, Perelman School of Medicine, Philadelphia, PA 19104, USA. [8]Present address: Department of Gynecology, Medical Faculty Mannheim of the Heidelberg University, Mannheim, Germany. [9]Present address: University of Pittsburgh, Hillman Cancer Center, Pittsburgh, PA, USA. ✉email: rdrapkin@pennmedicine.upenn.edu

Ovarian cancer, the most lethal gynecologic malignancy in developed countries, is a heterogeneous disease with multiple histologic subtypes that accounts for nearly 300,000 new cases and over 150,000 deaths worldwide each year[1–4]. The most common subtype of ovarian cancer is high-grade serous carcinoma (HGSC) and most patients with HGSC eventually develop recurrent disease that is resistant to cytotoxic chemotherapy. The poor survival rates are in part due to the lack of early detection tools and late clinical diagnosis[5,6]. Despite these dire statistics, significant progress has been made in recent years in our understanding of the pathogenesis of HGSC. Based on histologic, molecular, and genetic evidence, it is now generally accepted that a majority of HGSCs arise in the secretory cells of the distal fallopian tube (FT)[7–13]. More recently, next generation sequencing of fallopian tube precursors showed that mutations in *TP53* are among the earliest genetic events, occurring in benign appearing secretory cells that are non-proliferative. Stretches of these *TP53* mutated secretory cells are called 'p53 signatures' and are considered benign in isolation[14,15]. The acquisition of malignant cytological features and proliferation with expansion of these secretory cells leads to the formation of Serous Tubal Intraepithelial Carcinomas (STICs). Using whole-exome sequencing and mathematical modeling, two independent studies recently showed that the average time between development of a STIC lesion and ovarian cancer is approximately 6.5 years[9,13]. Unfortunately, once the malignant FT cells encamp on the surface of the ovary, seeding of the peritoneal cavity appears to occur rapidly thereafter. These data suggest that the ability of malignant FTSECs to disseminate and metastasize to the ovary is a critical event during the development of HGSC. It is currently unknown which molecular events trigger FT lesions to metastasize to the ovary.

The L1 cell adhesion molecule (L1CAM) is a type-1 transmembrane molecule that is over-expressed in various types of human cancers, including HGSC and may thus play an important role in neoplastic processes[16,17]. Many studies have shown that the expression of L1CAM is associated with malignant characteristics such as chemoresistance, epithelial to mesenchymal transition (EMT), proliferation, migration, invasion and survival[16]. While it is known that L1CAM is expressed in HGSC[16,17], it is not known at what stage of ovarian cancer development L1CAM is expressed and whether or not it has a functional role in transformation or dissemination.

In this report, we provide evidence that L1CAM plays a critical role in transformation and dissemination of HGSC FT precursors to the ovary. We found that L1CAM was highly expressed in the apical cells of STIC lesions and that its expression contributed to the dissemination of cells. L1CAM promoted FTSEC sphere formation and survival of ovarian cancer cells and immortalized FT cells. L1CAM mediated this effect through the activation of the integrin, AKT, and ERK signaling pathways which increased cell survival under substrate detachment conditions. Furthermore, utilization of a 3D ex-vivo model of human FTSEC spheres and mouse ovaries revealed that L1CAM was important for FT cells to invade the ovary as a cohesive group. Our results provide important mechanistic insight into the early steps of ovarian cancer metastasis from the fallopian tube to the ovary.

## Results

### L1CAM expression in high-grade serous carcinoma of the ovary.
Cell adhesion molecules modulate cell interaction with the extracellular environment, including other cell types and the extracellular matrix (ECM)[16]. L1CAM has been shown to have a critical role in metastatic progression in multiple tumor types including glioblastoma, melanoma, breast, renal, and colorectal cancers[18–22]. To address whether L1CAM plays a similar role in ovarian cancer, we examined the expression levels of L1CAM in HGSC by analyzing L1CAM RNAseq expression from The Cancer Genome Atlas (TCGA) data set. We first compared L1CAM expression in HGSC to other cancers from the Pan-Cancer cohort, a dataset containing over 10,000 different tumor samples (Fig. 1a)[23]. Our analysis revealed that HGSC exhibits some of the highest levels of L1CAM among solid tumors[24]. We found that L1CAM expression correlates with a shorter overall survival (Fig. 1b) and exhibits a gradual increase from Stage II to Stage IV disease (Fig. 1c). Interestingly, the biggest increase occurred between Stage IIIA and Stage IIIC. These stages differ by the degree of metastatic spread beyond the pelvis to the peritoneal cavity or omentum. Additionally, we found that L1CAM expression was significantly higher in patients with progressive disease compared to those with stable disease (Fig. 1d). When analyzing L1CAM expression in the four different molecular subgroups described in the original TCGA study (immunoreactive, proliferative, mesenchymal and differentiated), we found the highest L1CAM expression in the mesenchymal and differentiated fallopian tube-like subgroups; notably the mesenchymal subgroup displays the worst prognosis (Fig. 1e)[24]. Overall, these data are consistent with the reported role of L1CAM in other solid tumors and indicate that L1CAM expression correlates with ovarian cancer progression. However, whether L1CAM plays a role in early tumor development is unknown.

### Loss of RNF20 and H2B monoubiquitylation leads to L1CAM upregulation.
We recently reported that monoubiquitylation of histone H2B (H2Bub1) is an epigenetic post-translational mark that is lost early during HGSC development from the FT[25]. Loss of H2Bub1 alters chromatin accessibility and activates key pathways that facilitate progression of ovarian cancer. In mammalian cells, H2Bub1 levels are regulated primarily by the RNF20 E3 ligase[26]. To define the specific pathways impacted by loss of H2Bub1, we used two different shRNA constructs targeting the RNF20 transcript to knockdown *RNF20* gene expression in two immortalized FT cell lines, FT190 and FT194. Biochemical validation using Western blot analysis showed that depletion of RNF20 resulted in a pronounced reduction in H2Bub1 (Fig. 2a). RNAseq enrichment analysis revealed that the knockdown of RNF20 and H2Bub1 resulted in the up-regulation of seven common cellular pathways (Fig. 2b, Supplementary data 1, 2). The seven pathways were enriched in processes related to cell motility, extracellular matrix, and cell adhesion (Fig. 2c, Supplementary data 2). Common to the three pathways related to cell adhesion was L1CAM. In fact, reverse phase protein array (RPPA) analysis revealed that L1CAM was among the most significantly upregulated proteins upon RNF20 and H2Bub1 knockdown (Fig. 2d). Western blot analysis showed that depletion of RNF20 resulted in a marked increase in L1CAM in both cell lines (Fig. 2e). Given the early timing of H2Bub1 loss and the connection between H2Bub1, RNF20 and L1CAM, we hypothesized that L1CAM induction occurs in early FT precursor lesions.

### L1CAM expression in early HGSC precursor lesions.
To test the prediction that L1CAM is expressed in early HGSC development, we analyzed L1CAM protein expression by immunohistochemistry (IHC) in FT precursor lesions. IHC for p53, p16, and stathmin was used to credential seventeen cases of FT specimens with STIC lesions as previously described[25,27,28]. Using an L1CAM specific antibody[29,30], we found that L1CAM expression was readily detectable in 16 of the 17 STIC lesions (Fig. 3a, b). Interestingly, L1CAM expression was present in p53 negative (Fig. 3a) and p53 positive (Fig. 3b) precursors. We did not observe homogeneous L1CAM expression throughout the STIC

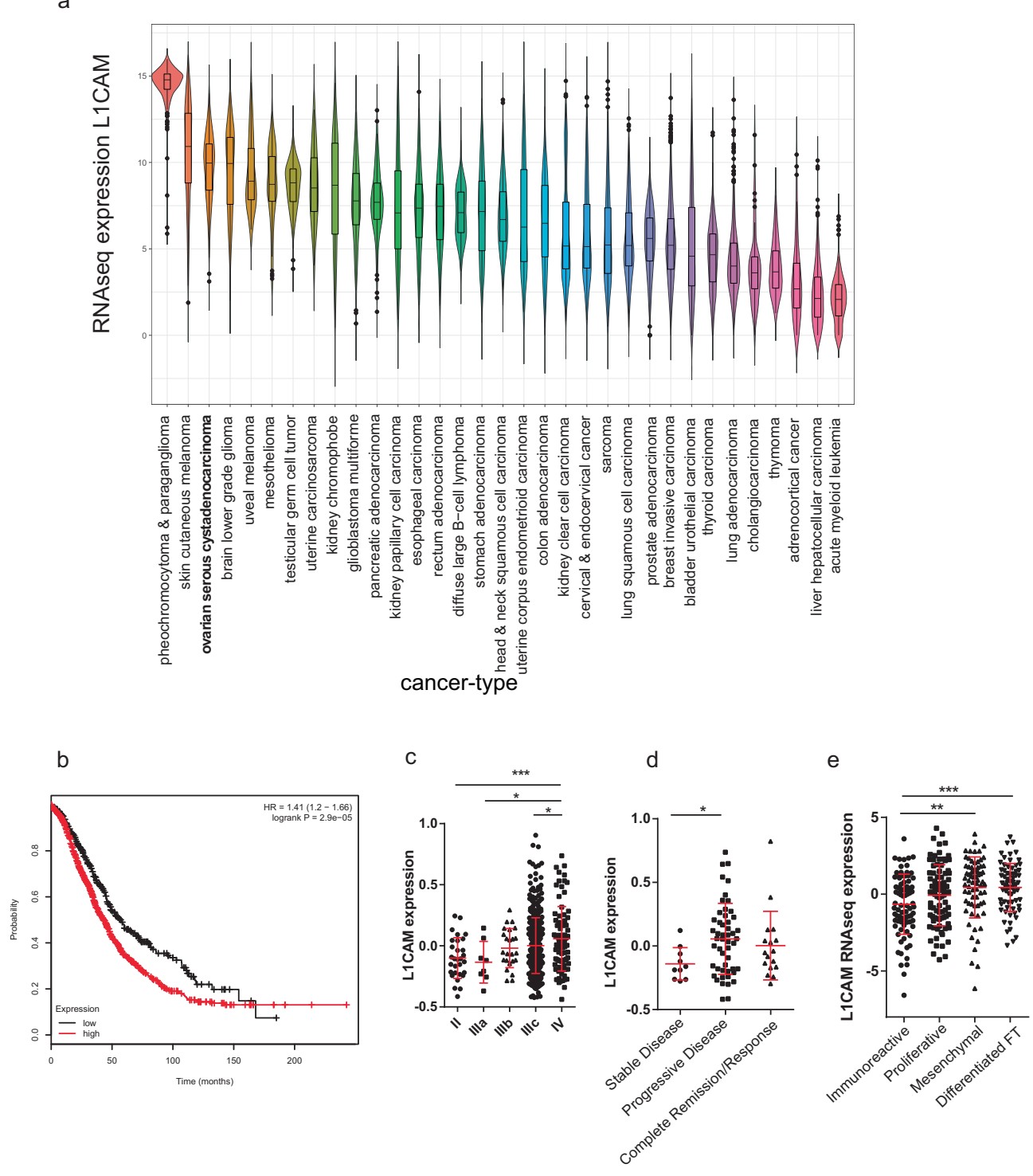

**Fig. 1 L1CAM expression in the TCGA ovarian cancer cohort. a** Violin plot of L1CAM expression in different cancer types from the PAN-Cancer cohort. **b** Kaplan–Meier overall survival analysis with the KM-Plotter of L1CAM (red line) against rest (black line). Cutoff value of 144 was used with the probe set 204584_at. $n = 1656$, $HR = 1.44$, $p = 6.1 \times 10^{-6}$. **c** L1CAM mRNA expression in tumors of the TCGA ovarian cancer cohort in different histological stages. **d**, **e** The same samples were analyzed for the relationship of L1CAM expression to disease progression and TCGA cluster, respectively. *P*-values were calculated with an unpaired two-sided *t*-test. $^{*}p < 0.05$, $^{**}p < 0.01$, $^{***}p < 0.001$.

lesions, but rather a gradient with peak expression near the apical regions of the STIC lesions (Fig. 3a–c). Some of the cells that stained strongest for L1CAM also appeared to be detached from the main STIC lesion, suggesting that L1CAM is enriched in disseminating STIC lesions[31]. In support of this hypothesis, in one section we found p53- and L1CAM-positive cell clusters that were detached from the original lesion and detected in an adjacent region of normal fallopian epithelium (Fig. 3d). In 10 benign FT epithelial sections, we observed only rare L1CAM-positive cells (Fig. 3d, e). Together, these data motivated us to further investigate the contribution of L1CAM to the phenotypes associated with early HGSC dissemination from the FT.

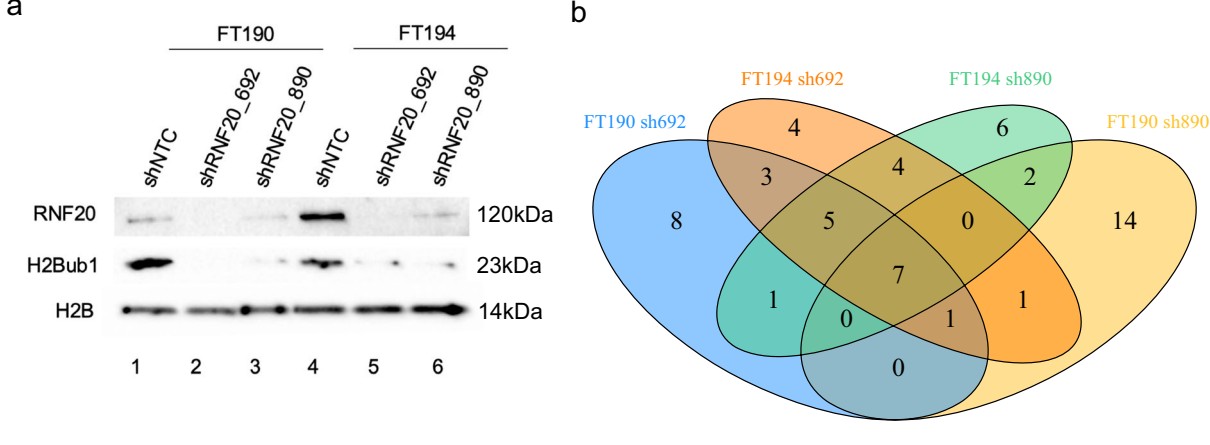

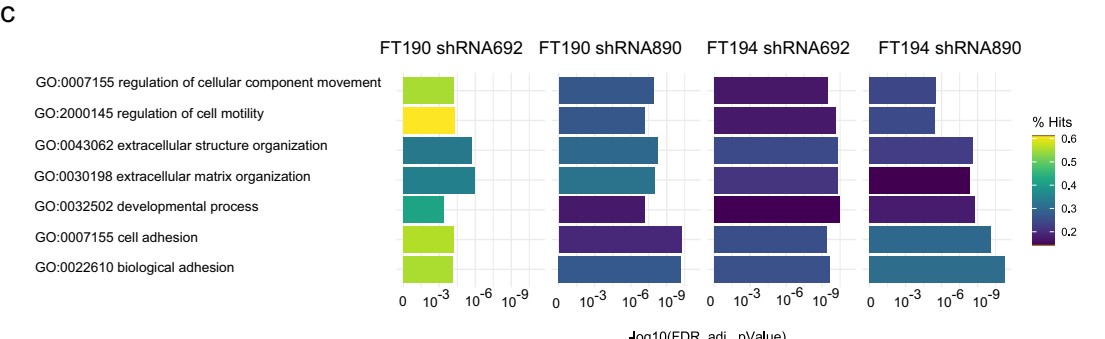

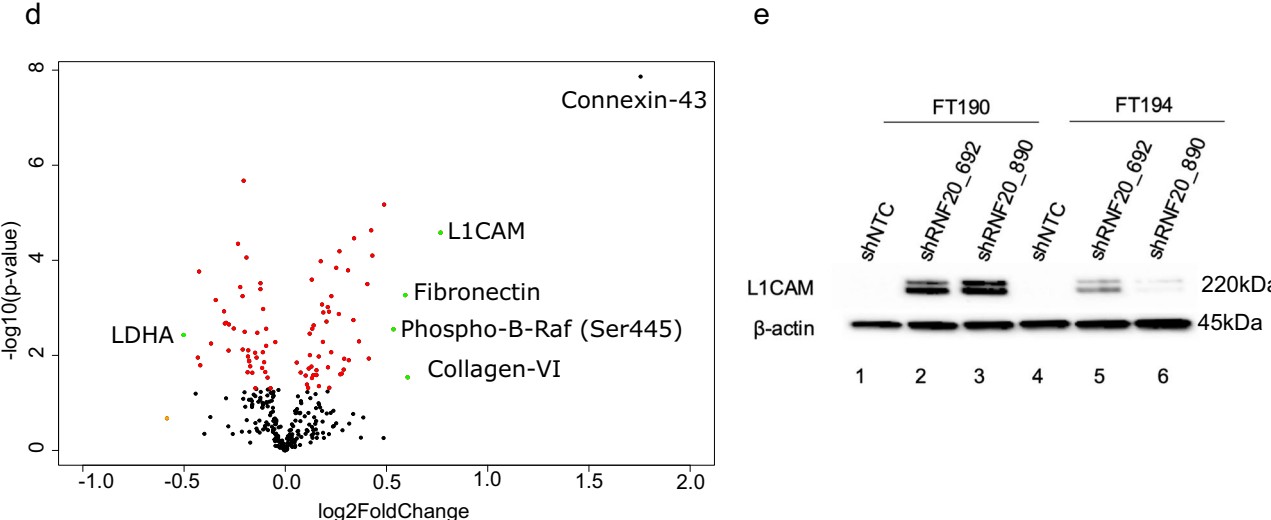

**Fig. 2 Loss of RNF20 leads to the upregulation of L1CAM. a** Western blot analysis of RNF20, H2Bub1, and H2B in FT190 and FT194 treated with two shRNAs targeting RNF20 or a control shRNA. **b** Venn diagram of the 25 most enriched pathways in FT190 and FT194 cells that were treated with two shRNAs (sh890 or sh692) to knockdown the RNF20 E3 ligase. **c** Display of the 7 pathways that overlap with all conditions in A. Bar graphs represent the False Discovery Rate (FDR) adjusted p-value in -log10 scale. The color represents percentage of regulated genes in each pathway. **d** Volcano plot of FT190 cells treated with RNF20 shRNA and analyzed with RPPA. Red dots depict the proteins that are significantly regulated. Orange dots depict samples that are regulated over 0.5 log2 fold (1.4 fold) and green depicts proteins that are significantly and 0.5 log2 fold regulated. **e** Western blot analysis of L1CAM and beta Actin in FT190 and FT194 treated with two shRNAs targeting RNF20 or a control shRNA.

**L1CAM expression in fallopian tube secretory cells and ovarian cancer cell lines.** To begin deciphering the role of L1CAM in HGSC, we analyzed L1CAM expression in a panel of ovarian cancer cell lines by Western blot and found that all of the cell lines expressed detectable L1CAM protein (Fig. 4a). Similarly, in primary cells derived from ascites fluid of HGSC patients, we found high L1CAM expression in seven of nine samples (Fig. 4b)[32]. In primary FTSECs that were derived from healthy donors and immortalized using different techniques[27,33,34], we found that some cell lines displayed little or no expression (FT33,

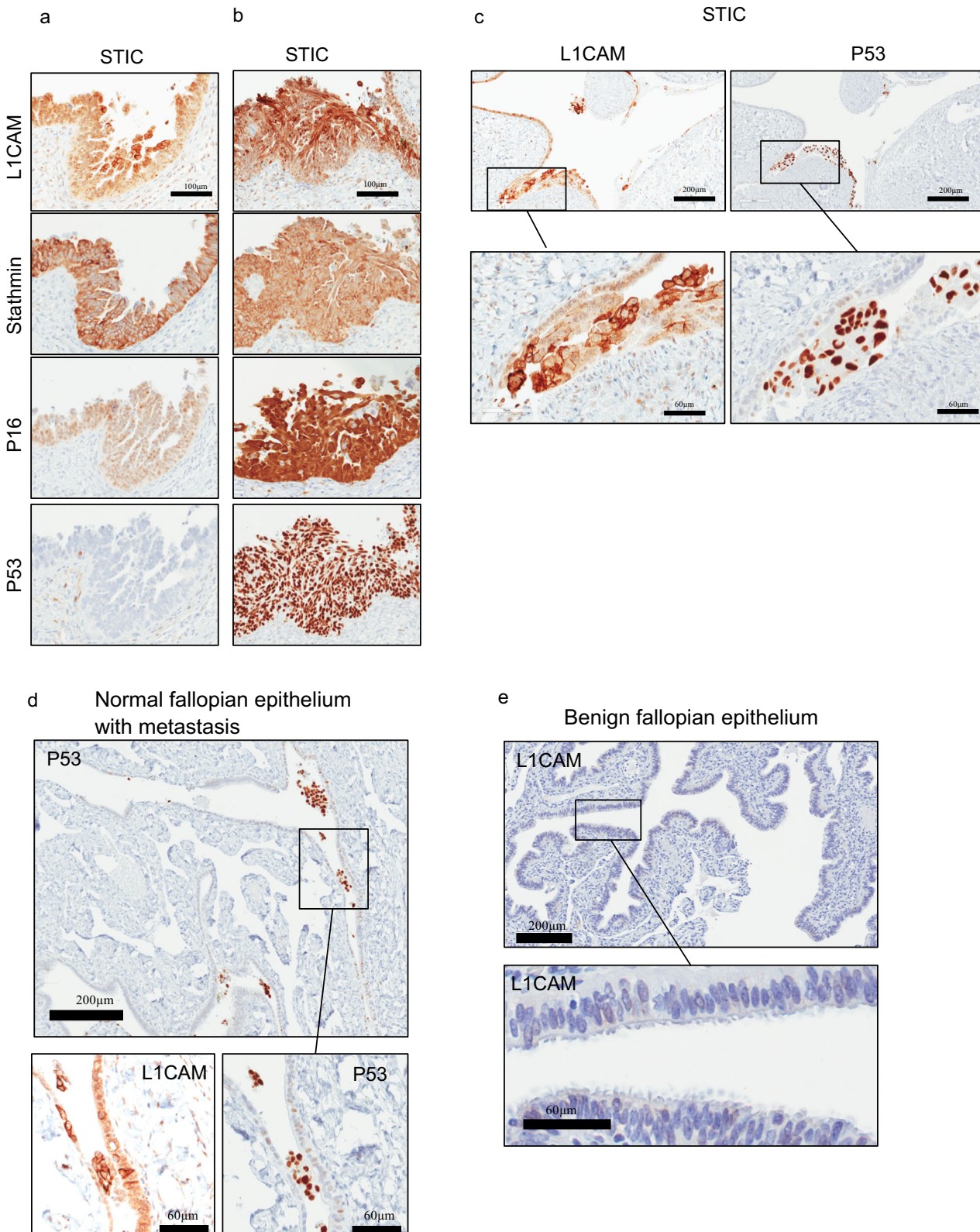

**Fig. 3 L1CAM expression in STIC lesions. a** p53-negative STIC lesion and **b** p53-positive STIC lesion. Both were stained for L1CAM, Stathmin, p16, and p53. **c** STIC lesion stained for L1CAM and p53. **d** Normal fallopian tube tissue with p53-positive metastatic cells floating in the tubal lumen stained for L1CAM and p53. **e** Normal fallopian tube tissue stained for L1CAM.

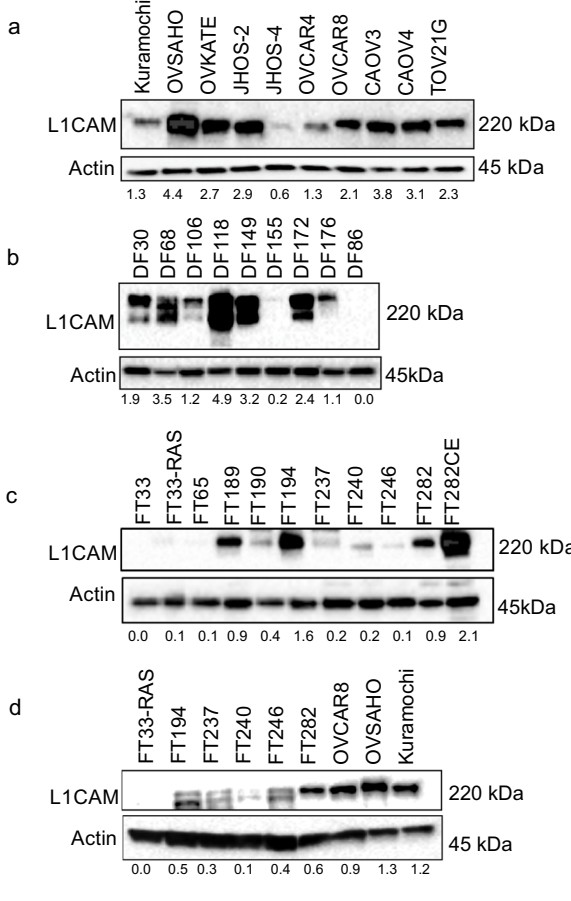

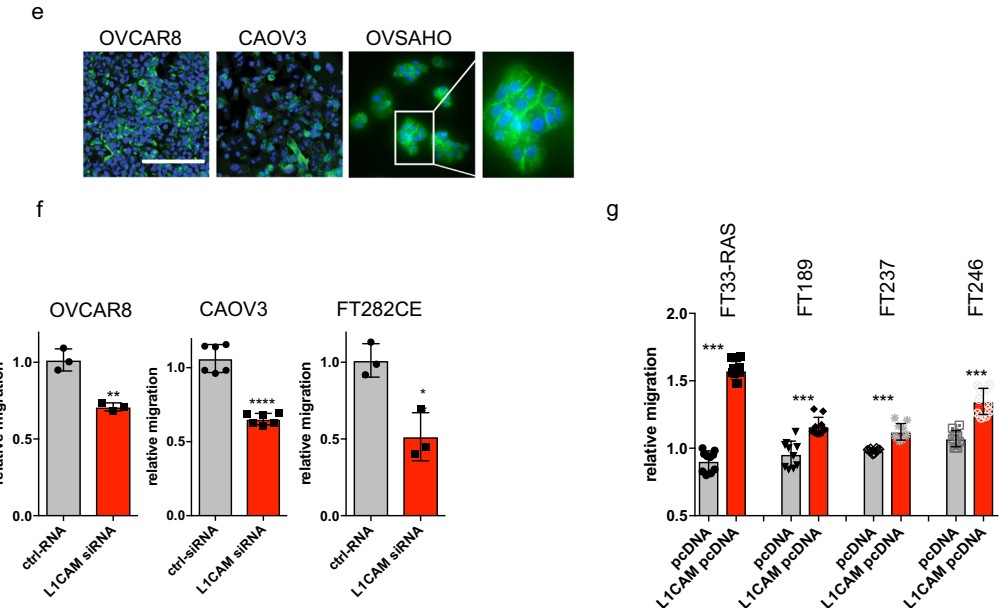

**Fig. 4 L1CAM expression in Ovarian cancer and primary cell lines.** Western blot analysis of L1CAM expression in ovarian cancer cell lines (**a**) primary ascites-derived HGSC cells (**b**) and immortalized fallopian tube cell lines (**c**) and a panel comparing immortalized fallopian tube cell lines to ovarian cancer cell lines (**d**). Actin was used as a loading control. Relative L1CAM expression was calculated by the ratio of the densitometry of L1CAM divided by Actin. **e** Immunofluorescence images of selected cell lines for surface L1CAM (green) and DAPI (blue) to stain the nucleus. **f** Relative migration in OVCAR8, CAOV3 and FT282-CE after siRNA knock-down of L1CAM or treatment with control siRNA. **g** Relative migration in FT33-RAS, FT189, FT237 and FT246, measured after overexpression of L1CAM or an empty control vector. **e** Scalebar represents 100 μm. **f, g** Three independent experiments were performed and P-values were calculated with an unpaired two-sided t-test. $^{*}p < 0.05$, $^{**}p < 0.01$, $^{***}p < 0.001$, $^{****}p < 0.0001$.

FT33-RAS, FT65, FT190, FT237, FT240, FT246), some showed moderate expression (FT189, FT194, and FT282) and one showed high expression (FT282CE) of L1CAM (Fig. 4c). This latter cell line was engineered to overexpress cyclin E1 and is partially transformed[34]. When directly comparing immortalized FTSECs with ovarian cancer cell lines, we found that L1CAM was more highly expressed in the cancer cell lines (Fig. 4d). While the immunofluorescence for surface bound membranous L1CAM correlated broadly with our Western blot data, it also showed that within individual cell lines, cells express a broad spectrum of L1CAM (Fig. 4e). The heterogeneity of L1CAM expression within cell lines, which was also observed by flow cytometry using surface staining of L1CAM, reflects our observation in STIC lesions (Supplementary Fig. 1a).

**L1CAM contributes to FTSEC migration**. To examine whether L1CAM contributes to phenotypes associated with tumor dissemination and metastasis, we used RNA interference to evaluate the impact of L1CAM knockdown on the migration of OVCAR8, CAOV3 and FT282CE cells (Supplementary Fig. 1b). We observed a significant reduction in migration in all three lines after L1CAM knockdown (Fig. 4f). Conversely, when we overexpressed L1CAM in fallopian tube cell lines with low or moderate expression of L1CAM (Supplementary Fig. 1c), we found a significant increase in cell migration (Fig. 4g). These data support the hypothesis that L1CAM is functionally involved in the regulation of ovarian cancer dissemination from the fallopian tube.

**Low L1CAM expression inhibits sphere formation in OVCAR8 cells**. Another important hallmark of ovarian cancer progression is the ability of tumor cells to detach from the primary site and transition into free floating multicellular structures that are supported by cell-cell and cell-matrix adhesion[35]. Since L1CAM is a cell adhesion molecule, we examined the contribution of L1CAM to the formation and maintenance of multicellular structures using the ovarian cancer cell line OVCAR8. We chose OVCAR8 cells because they spontaneously form three dimensional multicellular clusters when cultured under adherent conditions. We also observed by flow cytometry and immunofluorescence that L1CAM is heterogeneously expressed within multiple cells lines including OVCAR8 (Fig. 4e, Supplementary Fig. 1a). Using magnetic beads coupled to a L1CAM antibody we were able to negatively select and isolate isogenic OVCAR8 cells expressing undetectable L1CAM protein levels (Fig. 5a). The cells remained negative for L1CAM over multiple passages (up to 10 passages). The L1CAM-low OVCAR8 cells (termed as OVCAR8-L1$^{low}$) were significantly reduced in their ability to form compacted multicellular clusters under adherent culture conditions (Fig. 5b, Supplementary Fig. 2a) and under ultra-low adhesion conditions (ULA) (Fig. 5c). The reduced ability to form spheres under 3D conditions was most pronounced after 1 and 2 days, but was less dramatic after six days in culture (Fig. 5c, Supplementary Fig. 2b), indicating an important role for L1CAM in the initiation of sphere formation. In addition, we found that the loss of L1CAM strongly reduced colony formation, compaction and impeded the ability of the cells to invade (Fig. 5d–f, respectively). To confirm our findings, we used CRISPR/Cas9 to knockout L1CAM in OVCAR8 (Supplementary Fig. 2c, d). Similar to the L1CAM-low OVCAR8 cells, we observed a reduced ability to form multicellular aggregates in multiple L1CAM knockout clones (Supplementary Fig. 2e).

Importantly, when grown under serum free conditions, the ability to form spheres was completely abrogated in the OVCAR8-L1$^{low}$ cells (Fig. 5g) leading to an increase in cell death as measured by uptake of ethidium bromide in those

cultures (Fig. 5h). Ectopic re-expression of L1CAM in OVCAR8-L1$^{low}$ partially rescued aggregate formation (Fig. 5i, j), while the application of an inhibitory antibody that blocks homophilic L1CAM interactions (clone L1-9.3) significantly decreased the size of the multicellular clusters (Fig. 5k)[29]. Taken together these results indicate that L1CAM expression in ovarian cancer cells promotes cell-cell adhesion that supports multicellular cluster formation.

**L1CAM promotes multicellular cluster formation and anchorage independent survival in FTSEC cells**. To examine whether L1CAM is sufficient to promote multicellular clusters, we ectopically overexpressed L1CAM or an empty vector control in the FTSEC cell lines FT237, FT240, FT246 and FT282 (Fig. 6a). When L1CAM-expressing FTSECs were cultured as monolayer or under ULA conditions, they appeared to form multicellular clusters (Fig. 6b, c and Supplementary Fig. 3a). In contrast, control cells maintained a monolayer or generated loosely attached multicellular clusters (Fig. 6b, c and Supplementary Fig. 3a). These data indicate that L1CAM through its extracellular domain mediates cell-cell adhesion that leads to multicellular cluster formation. The increase in multicellular structure formation led to a reduction in cell death and an increase in the number of living cells when FT237 and FT240 were cultured under ULA and serum-free conditions (Fig. 6d and Supplementary Fig. 3b, c, respectively). We observed similar results when two additional FT lines overexpress L1CAM versus empty vector control (Supplementary Fig. 3d). To address the question whether the extracellular domain of L1CAM contributed to cellular cluster formation, we again applied the L1CAM-specific antibody (L1-9.3) that inhibits homophilic binding[29,36]. When cultured in the presence of the antibody, sphere size was significantly reduced compared to the IgG control (Fig. 6e)[36]. Interestingly, the effect of the antibody was most pronounced during the early phase of cluster aggregation, suggesting early involvement of L1CAM in mediating cell-cell adhesion. Taken together these data indicate that expression of L1CAM in FTSEC is sufficient to evoke multicellular aggregation and promote anchorage independent survival, the phenotypes associated with early tumor cell dissemination from the fallopian tube.

**L1CAM-mediates homophylic cell-cell interactions**. The observation that L1CAM expression leads to the formation of multicellular aggregates, defined by robust cell compaction and adhesion, led us to hypothesize that due to the homophilic binding of L1CAM, cells expressing high levels of L1CAM might preferentially interact with other cells expressing L1CAM. To test this hypothesis, we overexpressed L1CAM in two different fallopian tube cell lines and co-cultured these cells at a ratio of 1:10 with non-transduced control cells under ULA conditions. To visualize the co-cultures, we labeled the L1CAM-overexpressing FT cells red and the control cells green with fluorescent dyes. Interestingly, similar to the expression in STIC lesions, the L1CAM overexpressing cells clustered together within the FT sphere (Supplementary Fig. 4a). When we co-cultured OVCAR8 with normal fallopian tube cell lines at a ratio of 1:10, the OVCAR8 cells clustered and attached as cell groups to the fallopian tube clusters, but did not intersperse into the fallopian tube cells (Supplementary Fig. 4b). However, when co-culturing the OVCAR8-L1$^{low}$ cells with the fallopian tube cells, we observed a stronger intermixing of both cell types (Supplementary Fig. 4b). These data indicate that L1CAM-expressing cells bind to other L1CAM-expressing cells preferentially through homophilic binding and less so to cells lacking L1CAM.

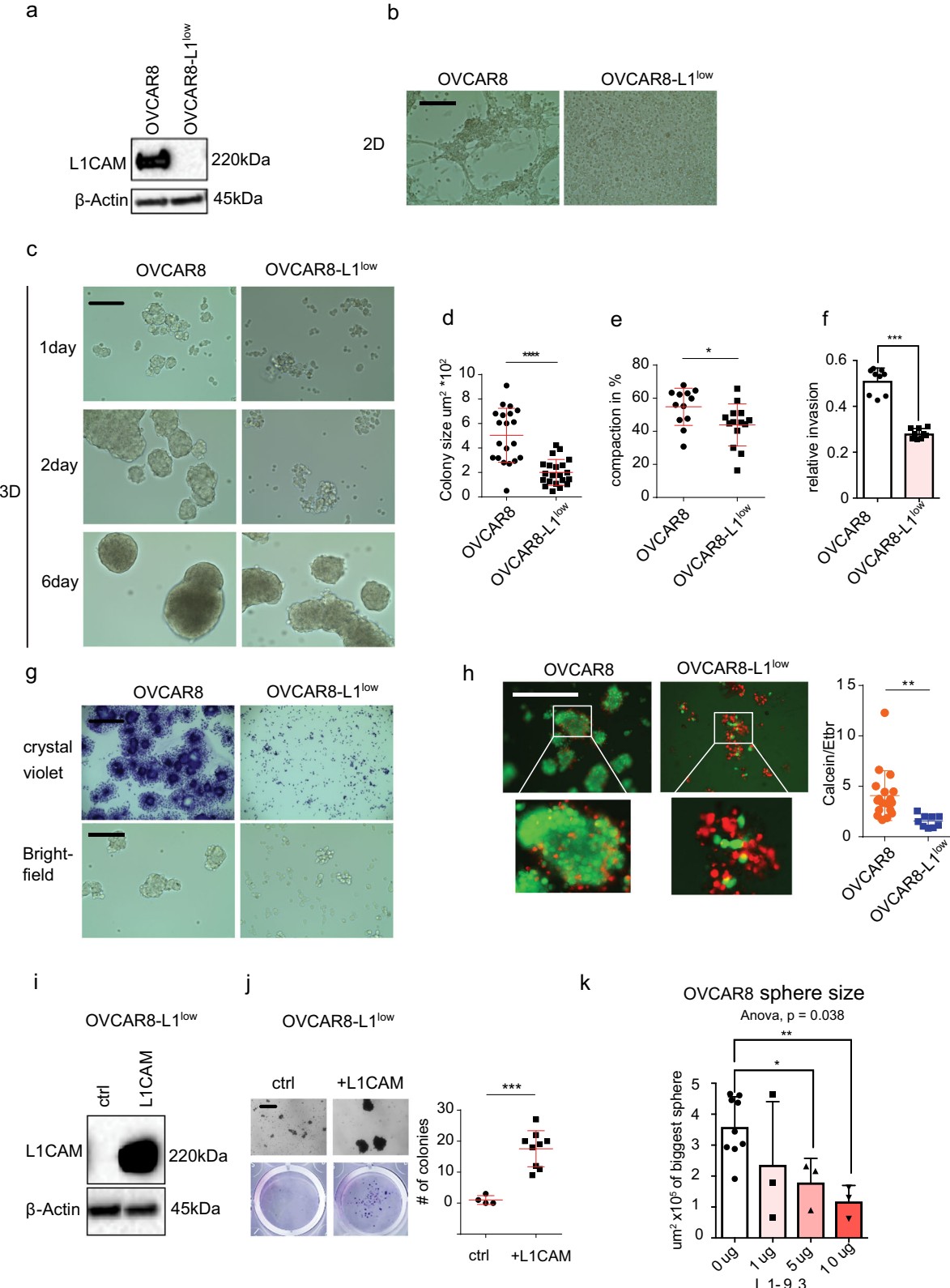

**L1CAM expression promotes extracellular matrix deposition and integrin expression in immortalized fallopian tube cells.** To examine L1CAM-dependent mechanisms associated with multicellular aggregate formation, we performed RPPA analysis (Supplementary Fig. 5) of control OVCAR8 and OVCAR8-L1low cells cultured as monolayers (2D) or multicellular clusters (3D). One of the most dramatic changes we noted was the near complete loss of fibronectin expression in the L1CAM-negative cells grown in 2D and 3D conditions (Fig. 7a, Supplementary Fig. 5a). We then analyzed fibronectin protein expression (Fig. 7b: RPPA-data, Y-axis) in the ovarian cancer TCGA dataset. We split the cases into three groups based on their L1CAM mRNA (X-axis) expression (high, medium and low; determined relative to median expression) and found that tumors expressing high levels of L1CAM also

**Fig. 5 L1CAM knockdown inhibits sphere formation in OVCAR8 cells. a** Western blot analysis of L1CAM expression in OVCAR8 and isogenic OVCAR8-L1low cells (passage 10). **b** Phase contrast images representing OVCAR8 and OVCAR8-L1low cells grown for 6 days under 2D conditions. **c** Phase contrast images representing OVCAR8 and OVCAR8-L1low cells grown for 1, 2 and 6 days grown under 3D conditions and for 6 days. **d** Colony formation assay was quantified for colony size in different OVCAR8 cell populations **e** Cells were measured for compaction (area change). **f** Relative invasion of OVCAR8 and OVCAR8-L1low cells. **g** OVCAR8 and OVCAR8-L1low cells analyzed by bright field microscopy (lower row) and for 3D colonies with crystal violet (upper row) after culturing under 3D conditions in serum free media. (**h**, left panel) Fluorescent images representing OVCAR8 and OVCAR8-L1low stained for calcein (green) and ethidium bromide (red). (**g**, right panel) Dot plot showing the ratio between living and dead cells, assessed by measuring the ratio between calcein (green) and ethidium bromide (red) intensity in OVCAR8 and OVCAR8-L1low cells after culturing for 7 days under 3D and serum free conditions. **i, j** Rescue experiment: Re-expression of L1CAM analyzed by Western blot (**i**) in OVCAR8-L1low and its effect on sphere formation. **j** Dot plot representing number of colonies after cells were transferred from 3D to 2D cell culture. **k** Effects of L1CAM antibody treatment (anti-L1- 9.3) on sphere size after 48 h. **b, c, g, h, j** Scalebar represents 200 μm. Three independent experiments were performed (**d, e, f, j**). *P*-values were calculated with an unpaired two-sided *t*-test. $^*p < 0.05$, $^{**}p < 0.01$, $^{***}p < 0.001$, $^{****}p < 0.0001$. **k** ANOVA and Dunnett multiple comparison tests were used to calculate significance.

expressed significantly more fibronectin protein (Y-axis) (Fig. 7b). To analyze the co-expression of L1CAM and fibronectin under 3D and 2D culture conditions, we re-seeded the spheres overnight on a coverslip and analyzed the attached spheres by immuno-fluorescence for fibronectin and L1CAM. Interestingly, we found that L1CAM and fibronectin showed the highest co-expression at the unattached portion of the sphere, while cells that attached to the plastic surface grew in a monolayer and showed a much lower expression (Fig. 7c).

Previous studies have documented the important roles of fibronectin and integrin-α5β1 in the establishment of extracellular matrix support for ovarian cancer cells[37,38]. Consistent with these functions, we observed a strong downregulation of integrin α5β1 (consisting of subunits α5 and β1) in the OVCAR8-L1low cells compared to control OVCAR8 cells (Fig. 7a). We next measured mRNA expression of fibronectin and integrin-α5 in control OVCAR8 and OVCAR8-L1low (Fig. 7d). We observed that L1CAM loss in OVCAR8 cells led to significant reduction of fibronectin and integrin-α5 mRNA expression levels (Fig. 7d). In addition, analysis of the Pan-Cancer cohort of the TCGA database confirmed a significant positive correlation of L1CAM expression with fibronectin and integrin-α5 (17 of 33 and 19 of 33 cancer types, respectively) expression in the majority of cancer types, including ovarian cancer (Supplementary Fig. 6a, b).

Based on these results, we next analyzed whether over-expression of L1CAM in primary FT cell lines also regulates the expression of fibronectin and integrin-α5β1 (Fig. 7e). While all FT cell lines already expressed high levels of fibronectin, L1CAM overexpression caused further increase in fibronectin expression, most notably in FT237. Additionally, we observed an increased expression of the integrin-α5 subunit in all the L1CAM-overexpressing cell lines. Taken together these data support the hypothesis that L1CAM through the regulation of integrin and matrix expression promotes multicellular structure formation and survival of neoplastic cells.

**Overexpression of L1CAM activates the ERK and AKT pathway in fallopian tube cells.** Our data indicate that L1CAM is required for the expression of integrin α5β1 and fibronectin; cell adhesion molecules that have been implicated in ovarian cancer cell survival and dissemination[18,20,22,39–41]. In addition, L1CAM expression has been reported to activate mitogen-activated protein kinase (MAPK) pathways in several cell line models. Thus, we hypothesized that L1CAM is involved in the regulation of pro-survival signaling pathways in FTSEC and cancer cell lines. To test our hypothesis, we analyzed a MAPK and AKT signaling array after overexpression of L1CAM in two immortalized FT cell lines. We found a strong activation of ERK and AKT in the L1CAM-expressing cells compared to controls (Fig. 7e, f, Supplementary Fig. 7a). Correspondingly, we observed reduced activity in both pathways in the OVCAR8-L1low cells

(Supplementary Fig. 7b), which is also in agreement with previous reports, though our results did not reach statistical significance[42–44]. Furthermore, siRNA-mediated attenuation of fibronectin or integrin-α5 expression abrogated the activation of ERK and AKT in the fallopian tube cell lines, while it decreased AKT but not ERK activation in OVCAR8 cells (Fig. 7g, Supplementary Fig. 7c–e). Importantly, the knockdown of fibronectin or integrin-α5 also reduced the ability of OVCAR8 and FT237 cells to form cell aggregates (Fig. 7h, Supplementary Fig. 7c–e), suggesting that cell-matrix-cell adhesion mediated by integrin α5 supports tumor cell survival. These results are consistent with the role of integrins and ECM in supporting ovarian cancer dissemination.

**L1CAM is sufficient to support ovarian cancer colonization to the ovary.** Our results support the hypothesis that L1CAM promotes cell migration with formation and survival of multicellular aggregates, two phenotypes associated with early ovarian cancer dissemination from the fallopian tube to the ovaries. L1CAM mediates these activities through the stabilization of integrin-matrix complexes that activate pro-survival pathways including AKT and ERK. To test whether ovarian cancer cells and L1CAM-expressing FTSECs can colonize the ovary, we developed an ovary-tumor co-culture system (Fig. 8a). Our goal was to mimic the in vivo situation where the close proximity of the FT fimbria to the ovarian surface would increase the likelihood of early tubal HGSC precursors shedding, attaching to, and invading the ovary. The ovary appears to provide an ideal microenvironment and scaffold for cancer cells to grow and expand before metastasizing to other organs[35,45,46]. We labeled freshly isolated mouse ovaries with a green fluorescent dye and co-cultured the ovaries under ULA conditions with OVCAR8 or FTSEC that expressed either the red fluorescent protein (RFP) or were labeled with a red fluorescent dye. After 24 h of co-culture, we observed only a few cells of both OVCAR8 and OVCAR8-L1low attaching to the ovary. However, between 2–7 days of co-culture, the majority of OVCAR8 cells expressing L1CAM started to aggregate before attaching as cellular clusters to the ovary surface (Fig. 8a). These spheres then started to invade into the ovary (Fig. 8b). In contrast, in the OVCAR8-L1low - ovary co-culture, multicellular clusters attached to and invade the ovary to a lesser extent (Fig. 8b, c). We then analyzed the co-cultures by IHC. We confirmed that the spheres of OVCAR8 attached to and invaded into the ovary (Fig. 8d). Interestingly, we also detected invasion by OVCAR8-L1low cells but this was mostly restricted to single cells and only at sites on the ovarian surface that exhibited disruptions of the ovarian surface epithelium that were formed by manipulation of the ovaries (Fig. 8e). Furthermore, immunofluorescence of L1CAM, PAX8 and Cytokeratin 8 revealed that attached spheres of OVCAR8 cells spread both vertically below the OSE and laterally into the ovary tissue (Fig. 8f).

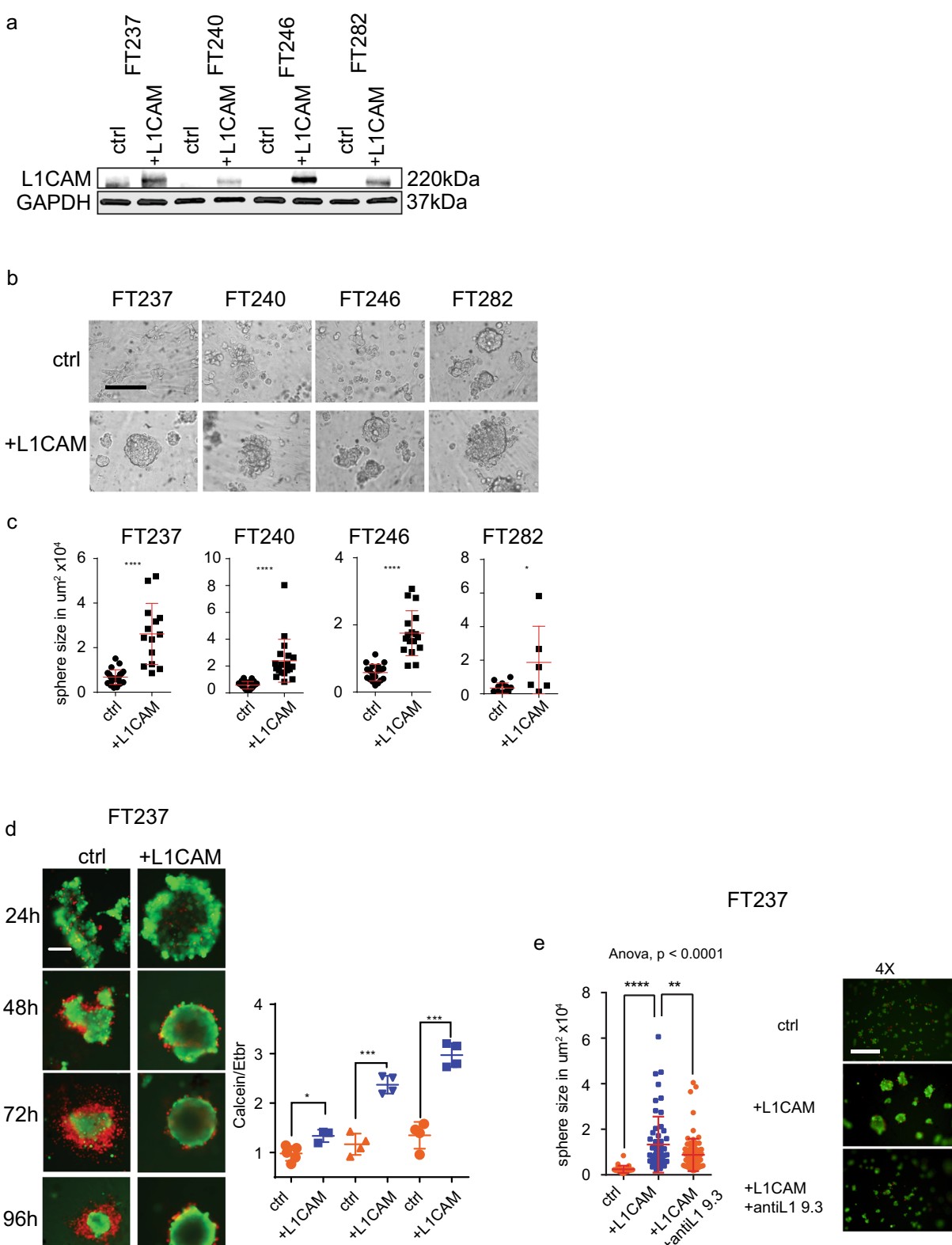

Finally, we overexpressed L1CAM in FT237 and FT246 cell lines to examine whether L1CAM might also influence the ability of these cells to adhere and invade the ovary. The co-culture of the FT cells with ovaries revealed an increased attachment and invasion of L1CAM-positive FT cells to the ovaries when compared to control cells, demonstrating the importance of L1CAM in metastatic attachment and invasion of HGSC precursors (Fig. 8g). Overall, our results support the model whereby fallopian tube precursor cells utilize L1CAM to regulate integrin expression and signaling to support ovarian cancer dissemination to the ovary.

**Fig. 6 L1CAM triggers sphere formation in primary fallopian tube cell lines. a** Western blot analysis of primary fallopian tube cell lines transfected with L1CAM or a control vector and analyzed for L1CAM and GAPDH (housekeeping gene) expression. **b** Bright field images of primary fallopian tube cell lines from a. Scalebar represents 200 μm. **c** Quantification of area in μm$^2$ of spheres shown in **b**. **d** (right panel) Fluorescent images representing FT237 cells transfected with a L1CAM or (left panel) a control vector. Dot plot depicting the ratio between living and dead cells, assessed by measuring the ratio between calcein (green) and ethidium bromide (red) intensity in FT237 cells transfected with L1CAM or a control vector after culturing under 3D and serum free conditions. Scalebar represents 100 μm. (**e**, left panel) Dot plot showing sphere size of FT237 cells that was analyzed after transfection with control vector or L1CAM with the addition of either IgG antibody or the anti L1-9.3 antibody. (**e**, right panel) Fluorescent (calcein, green) images representing FT237 cells transfected with control vector (left) or L1CAM with the addition of either IgG antibody or the anti L1-9.3 antibody. Scalebar represents 250 μm. Three independent experiments were performed (**c**, **d**). P-values were calculated with an unpaired two-sided t-test. $^*p < 0.05$, $^{**}p < 0.01$, $^{***}p < 0.001$, $^{****}p < 0.0001$. **e** ANOVA and Dunnett multiple comparison tests were used to calculate significance.

## Discussion

Recent molecular studies on the development of HGSC emphasize the need to better understand the evolution of precursor FT lesions and factors that trigger dissemination to the ovary and beyond[7,9,13,47]. Here, we report on the functional contributions of the L1CAM adhesion molecule to early HGSC progression from the fallopian tube. Using orthogonal approaches, we make three novel observations. First, using a combination of immunohistochemistry and transcriptomic studies we show that L1CAM is expressed in fallopian tube STIC lesions, particularly in cells that are detaching from the STIC. The induction of L1CAM expression in this setting is triggered by loss of RNF20 and H2Bub1 in early HGSC FT precursors. Second, we show that knockout of L1CAM partially abrogates the ability of cancer cells to form spheres. Conversely, overexpression of L1CAM in FT cells engenders sphere forming abilities on these otherwise benign secretory cells. Our proteomic studies show that L1CAM, through the regulation of integrin and matrix expression, promotes multicellular structure formation and survival of neoplastic cells. Finally, using an ovarian tissue explant model, we show that L1CAM-expressing cells are able to effectively colonize the ovary while L1CAM-depleted cells cannot.

Prior studies have shown that L1CAM, an essential cell adhesion molecule, is aberrantly expressed in tumor cells where it promotes cell migration and invasion[17,19,48–53]. Our analysis of TCGA data shows that HGSC exhibits among the highest levels of L1CAM relative to other solid tumors and that its expression is correlated to more aggressive disease and poor clinical outcomes. These findings are consistent with prior reports[17,52,53] but do not shed light on the role of L1CAM in early cancer pathogenesis. We recently reported that a particular epigenetic mark, monoubiquitylation of histone H2B (H2Bub1), is lost in FT precursors of HGSC[25]. Here, we show that loss of H2Bub1 triggered upregulation of L1CAM in these HGSC precursors. Using immunohistochemistry for L1CAM on human FT precursor lesions, we found that peak L1CAM expression was largely restricted to cells that appeared to be detaching from the precursor lesion. Therefore, we hypothesized that L1CAM might have a functional role during the early dissemination of cells from STIC lesions to the ovary.

During the transition from STIC lesions to HGSC, shedding to the ovary is thought to be a critical step[35,45]. Presumably, the ovary provides a rich microenvironment and scaffold for rapid growth, expansion, and eventual dissemination. Yang-Hartwich et al. (2014) showed that the high content of collagen IV and the chemokine SDF-1 (CXCL12) are key factors to attract and maintain cancer cells in the ovaries[35]. Additionally, using a mouse model, our lab showed that STIC lesions require spread to the ovary prior to peritoneal dissemination[45]. Importantly, the removal of ovaries in this mouse model resulted in a reduction of peritoneal metastasis, indicating the importance of the ovary as an optimal niche for cancer cells to grow and progress prior to becoming more widely metastatic. Cancer cells from STIC lesions have to master serial steps to successfully establish metastatic lesions in the ovary: 1) dissemination from the STIC lesion; 2) survival in the space between fallopian tube and ovary; 3) attachment to the ovary: 4) invasion into the ovary; and 5) establishing a self-sustained metastatic lesion.

We used a combination of approaches and models to characterize the role of L1CAM during these steps. We found that the ability of OVCAR8 cells to transition from a 2D monolayer to 3D spheres was abrogated by the loss of L1CAM. The loss of L1CAM also reduced the ability of cells to form spheres under ULA conditions which instead induced cell death. Conversely, overexpressing L1CAM in multiple normal FT cell lines led to an increase in sphere formation and reduced cell death under ULA conditions. These findings suggest that L1CAM enables survival of FT-derived cells under anchorage-independent conditions, thereby facilitating the dissemination step. The finding that L1CAM-specific antibodies were able to partly block the functions of L1CAM in vitro is consistent with animal studies showing that L1CAM-specific antibodies can block tumor progression in a peritoneal cell line xenograft model[29,54]. Further studies are needed to establish whether L1CAM can be a target in this setting.

By co-culturing L1CAM-positive and -negative fallopian tube cells under ULA conditions, we showed that L1CAM-positive cells preferentially attached to other L1CAM-expressing cells. This might simulate the situation in a STIC lesion, in which the strong interaction of L1CAM-positive cells leads to the coalescence and dissemination of those cells. Indeed, a number of studies have indicated that homotypic cluster formation can render cells resistant to anoikis[37]. Similarly, when co-culturing the OVCAR8 cells (expressing L1CAM) with FT cell lines under ULA conditions, the OVCAR8 cells assembled isolated colonies around the FT cells. In contrast, the OVCAR8-L1$^{low}$ cells showed no preference to bind to each other and intermixed with the FT cells. Importantly, we found that L1CAM expression leads to the upregulation of integrin-α5β1 and fibronectin under 3D conditions, which subsequently led to the activation of the ERK and AKT pathways. The upregulation of integrin-α5β1 has been shown to play an important role during tumorigenesis and in the promotion of metastasis in HGSC[55–58]. Integrin-α5β1 has also been implicated in sphere formation and activation of the AKT and ERK pathways[55,56]. The upregulation of integrin-α5β1 and fibronectin is important for the survival of cell aggregates that lose contact to their primary substratum[59].

Consistent with our data, the laboratory of Joanna Burdette has shown in a similar ex-vivo model that the loss of PTEN allows the growth of multicellular tumor spheroids under ultra-low adhesion conditions[60]. Importantly, like L1CAM overexpression, the loss of PTEN leads to the activation of the AKT pathway, indicating that the same signaling pathway is responsible for an increase in multicellular tumor spheroid formation and metastasis.

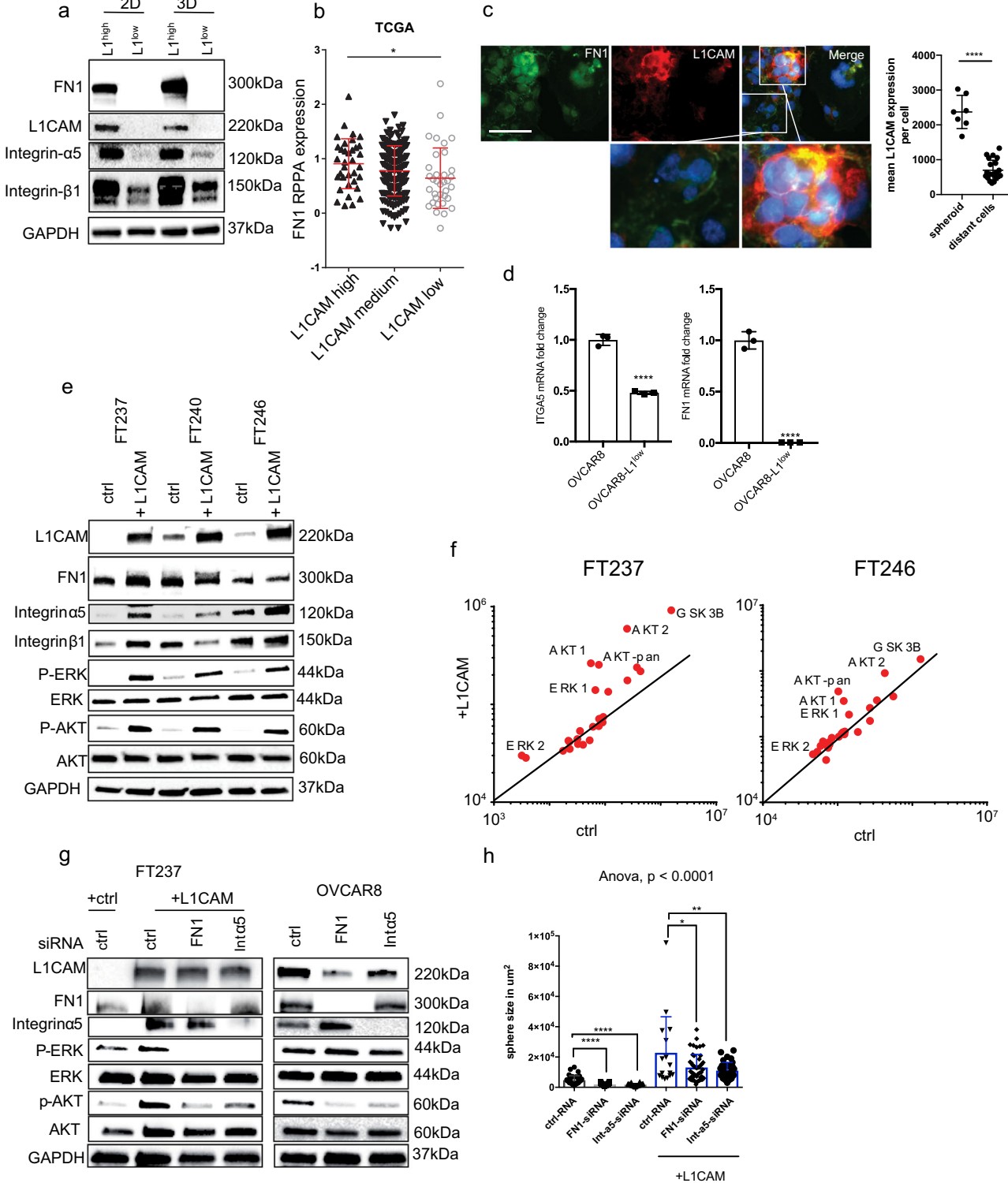

HGSC transgenic mouse models that include a PTEN loss have been described in multiple studies including a recent study that explores the development and progression of metastatic ovarian cancer[61]. Future studies in a similar transgenic model using overexpression of L1CAM, instead of PTEN loss, would further support the role of L1CAM in ovarian cancer dissemination.

For cancer cells to invade the ovary, cells have to detach from the FT, survive in the space between the FT and the ovary and attach to the OSE. To model and assess this process, we created an ex-vivo system in which we cultured isolated ovaries together

with cancer cells or fallopian tube cells under ULA culture conditions. Consistent with prior findings, we observed that after 24 h, some single cells attached to the ovarian surface at sites that showed a disruption of the OSE[35,46]. During this initial attachment stage, we did not observe differences between the L1CAM-positive or -negative cancer or FT cells. In general, we found that cells need to first coalesce into spheres before they can attach and invade the ovary. Interestingly, we observed that the spheres of L1CAM-positive cells were able to attach and invade through the OSE into the ovary stroma, whereas L1CAM-negative spheres

**Fig. 7 L1CAM regulates the expression of fibronectin during 3D sphere formation. a** Western blot analysis of fibronectin, Integrin α5, Integrin β1, Integrin β3, L1CAM and GAPDH in OVCAR8 and OVCAR8-L1[low] under 2D and 3D culturing conditions. Representative experiment ($n = 5$) is shown. **b** Dot plot representing fibronectin RPPA protein expression in ovarian cancer tumors of the TCGA cohort with a high, medium and low L1CAM mRNA expression. **c** Fluorescent images representing L1CAM (red) and fibronectin (green) in OVCAR8 cells cultured under 3D conditions before re-seeding on coverslips for fixation. Scalebar represents 200 μm. **d** mRNA fold change of Integrin alpha 5 (left) and Fibronectin (right) measured by quantitative real time PCR in OVCAR8 and OVCAR8-L1[low] cells. **e** Western blot analysis of FT237, FT240 and FT246 transfected with either L1CAM or a control vector. Representative Western blot ($n = 3$) is shown. **f** Scatter plot representing protein kinase array in FT237 and FT246 expressing control plasmid and plasmid-containing L1CAM. **g** Western blot analysis of FT237 and OVCAR8 analyzed 72 h after siRNA treatment, targeting fibronectin, Integrin α5. Representative Western blot ($n = 3$) is shown (**h**) Box plot representing quantification of sphere size of FT237-ctrl and FT237 + L1CAM after siRNA knockdown of fibronectin or Integrin α5. Three independent experiments were performed (**b**, **c**). P-values were calculated with an unpaired two-sided t-test. *$p < 0.05$, **$p < 0.01$, ***$p < 0.001$, ****$p < 0.0001$. **h** ANOVA and Dunnett multiple comparison tests were used to calculate significance.

were only able to attach weakly to the OSE and did not invade. Our data suggest that L1CAM enables survival of detached cells and promotes sphere formation that is a prerequisite for ovarian invasion. These L1CAM-dependent activities increase the likelihood of successfully establishing a metastatic lesion on the ovary that can facilitate the progression to HGSC.

In summary, we examined the role of L1CAM in the early dissemination and survival of HGSC precursors from the FT. We showed that L1CAM is upregulated in precursor tumor cells and that it is required for sphere formation and survival under anchorage-independent conditions. L1CAM mediated these activities through upregulation of integrin-α5β1 and fibronectin and activation of the AKT and ERK pathways. Our 3D ex-vivo model showed that L1CAM expression was important for FT cells to invade the ovary as a cohesive group. L1CAM upregulation appears to be related to the loss of H2Bub1 in HGSC precursors but further studies are required to identify the exact mechanism. The important role of L1CAM in the development and propagation of HGSC suggests that L1CAM may have potential as a target to prevent ovarian cancer dissemination.

## Methods

**Cell lines.** The establishment of the fallopian tube cell lines has been previously described[27,33,34]. They were cultured in Dulbecco's Modified Eagle's Medium (DMEM)/Ham's F-12 1:1 (Cellgro) supplemented with 2% Ultroser G serum substitute (Pall Life Sciences, Ann Arbor, MI, USA) and 1% penicillin/streptomycin. All cancer cell lines (see Supplementary data 3) were cultured in RPMI 1640 (Invitrogen, Carlsbad, CA) supplemented with 10% fetal bovine serum (FBS, Atlanta Biologicals) and 1% penicillin/streptomycin (Invitrogen). All cells were grown at 37 °C and a 5% $CO_2$-containing atmosphere.

**TCGA analysis.** Data from the TCGA database were extracted and downloaded from the XENA portal of the University of California, Santa Cruz (http://xena.ucsc.edu/). The extracted copy number and RNAseq data from the TCGA ovarian cancer cohort, TCGA pan-cancer cohort and the Cancer Cell Line Encyclopedia were analyzed with the GraphPad Prism or the ggplot2 package of the R software.

**Migration assay.** This assay was used to study the ability of cells to migrate through a coated transwell-plate placed (Corning) in a 24-well-plate. The bottom side (outside) of the polycarbonate membrane (with a pore size of 5 μm) was pre-coated with fibronectin (10 μg/ml) for 90 min at 37 °C (Supplementary data 4). The lower compartment was filled with 600 μL serum-free medium and the transwell insert was transferred into the well. The cells were diluted to $1 × 10^5$ cells in 100 μl in serum free medium and transferred to the upper compartment and incubated in a cell culture incubator for 16 h at 37 °C. Cells in the upper compartment were removed using a cotton swab. Migrated cells were stained with 500 μl of a crystal violet buffer (Crystal Violet 0.05% w/v, Formaldehyde 1%, 10X PBS (1X), Methanol 1%) for 45 min. The transwell was washed extensively in water before the microporous membrane was cut out and transferred to a new well containing 300 μl 10% acetic acid. The eluted dye was transferred to a 96-well-plate and the absorption was measured in a plate reader at 570 nm.

**Ultra-low attachment assay.** Cells were detached by 0.25% trypsin, resuspended in culturing media, centrifuged for 5 min at 1000 rpm and resuspended in serum free media. For the live/dead assay, $5 × 10^3$ cells were seeded per well of a 384-well ultra-low-adhesion (ULA) plate (Corning, catalog number: 4588) and 2 μM calcein, 3 μM of ethidium bromide (Supplementary data 4) was added to the cells and incubated for three hours. Cells were analyzed on a fluorescence microscope and

mean intensity of the green and red fluorescence channel was analyzed with imageJ.

**CRISPR-Cas9 Targeting of L1CAM.** To knockout the L1CAM gene in OVCAR8 cells, we used the Synthego CRISPR system and followed the manufacturers protocol. Briefly, we co-transfected Cas9 protein with 3 different sgRNAs by using the transfection reagent Lipofectamine CRISPRMAX.

Sequence 1: GGUGCCCAGCUUAUUGC
chromosome X, GRCh38.p13 153872186-153872202
Sequence 2: GCUGUUGUUGCCCGUGA
chromosome X, GRCh38.p13: 15387224-153872262
Sequence 3: CAAACCCAAGGAAGAGC
chromosome X, GRCh38.p13: 153872324 - 153872308

Cells were seeded in a 24-well plate 24 h prior to transfection to a confluency of approximately 80%. 3pmol of the Cas9 Nuclease and 3.9pmol of each sgRNA, were diluted in 25 μL Opti-MEM media together with 1 μL Lipofectamine Cas9 Plus reagent. 1.5 μL of the Lipofectamine CRISPRMAX transfection reagent diluted in 25 μL Opti-MEM media, were then mixed with the nucleic acid dilution. The nucleic acid/transfection reagent solution was incubated at room temperature for 5 min before being added to the cells. The cells were incubated for 72 h before seeding single cells into 96-well plates. Grown clones were expanded and analyzed by FACS and Western blot for the loss of L1CAM expression. Potential clones were sequenced to confirm cleavage. We used 4 different clones for this study (clones 1, 3, 11, 12).

**3D ovary invasion assay.** Cancer or fallopian tube cells, cultured in a 10 cm dish or flask, were detached with 0.25% trypsin, resuspended in culturing media, and centrifuged for 5 min at 1000 rpm. Cells not expressing a fluorescent marker were resuspended in serum free media and $1 × 10^5$ cells/ovary were incubated with a red fluorescent dye (CellTracker™ Red CMTPX, life technologies) for 30 min at 37 °C before washing 3 times with PBS.

Adult C57BL/6 female mice (10-14 weeks old) were sacrificed by using a $CO_2$ chamber before whole ovaries were harvested. The ovaries were surgically separated from fallopian tubes under a dissecting microscope and the bursa was removed. The ovaries were washed two times in PBS with 10% penicillin streptomycin on ice. Ovaries (2 to 4 ovaries per experimental condition) were then incubated with a green fluorescent dye (CellTracker™ Green CMFDA, life technologies) dissolved in serum free medium for 30 min at 37 °C before washing 3 times with PBS. Organs and $1 × 10^5$ cells were then added to 1 ml culturing media in an ultra-low attachment 24 well plate (Corning) and imaged every 24 h. for up to 2 weeks for red and green fluorescence. All animal protocols were approved by the Institutional Animal Care and Use Committee at the University of Pennsylvania.

**3D co-culture experiment.** Cancer or fallopian tube cells, cultured in a 10 cm dish or flask, were detached by 0.25% trypsin, resuspended in culturing media, and centrifuged for 5 min at 1000 rpm. Cells not expressing a fluorescent marker were resuspended in serum free media and incubated with a red or green fluorescent dye (CellTracker™ Red CMTPX, Green CMFDA, life technologies) for 30 min at 37 °C before washing 3 times with PBS. Marked cell were co-cultured in normal media and ULA plates (Corning) in a ratio of 1:10 of L1CAM expressing cells (red) to control cells (green), respectively.

**Compaction assay.** OVCAR8 cells were seeded in ULA 96-well round bottom plates (Corning). Cells were briefly spun at 127 g for 3 min. To generate cellular clusters, cells were returned to the incubator for 24 to 48 hours. Cellular clusters were then imaged for 48 h at 20- to 30 min intervals using a Nikon Ti-E inverted motorized widefield fluorescence microscope with integrated Perfect Focus System and (20×, 0.45 *numerical aperture* (NA)) magnification/NA phase/DIC optics equipped with a $CO_2$- and temperature-controlled incubation chamber[37].

**Colony formation assay.** 2000 cells were seeded on 6-well plates and cultured for two weeks. Cells were then washed with PBS and stained with the Crystal Violet

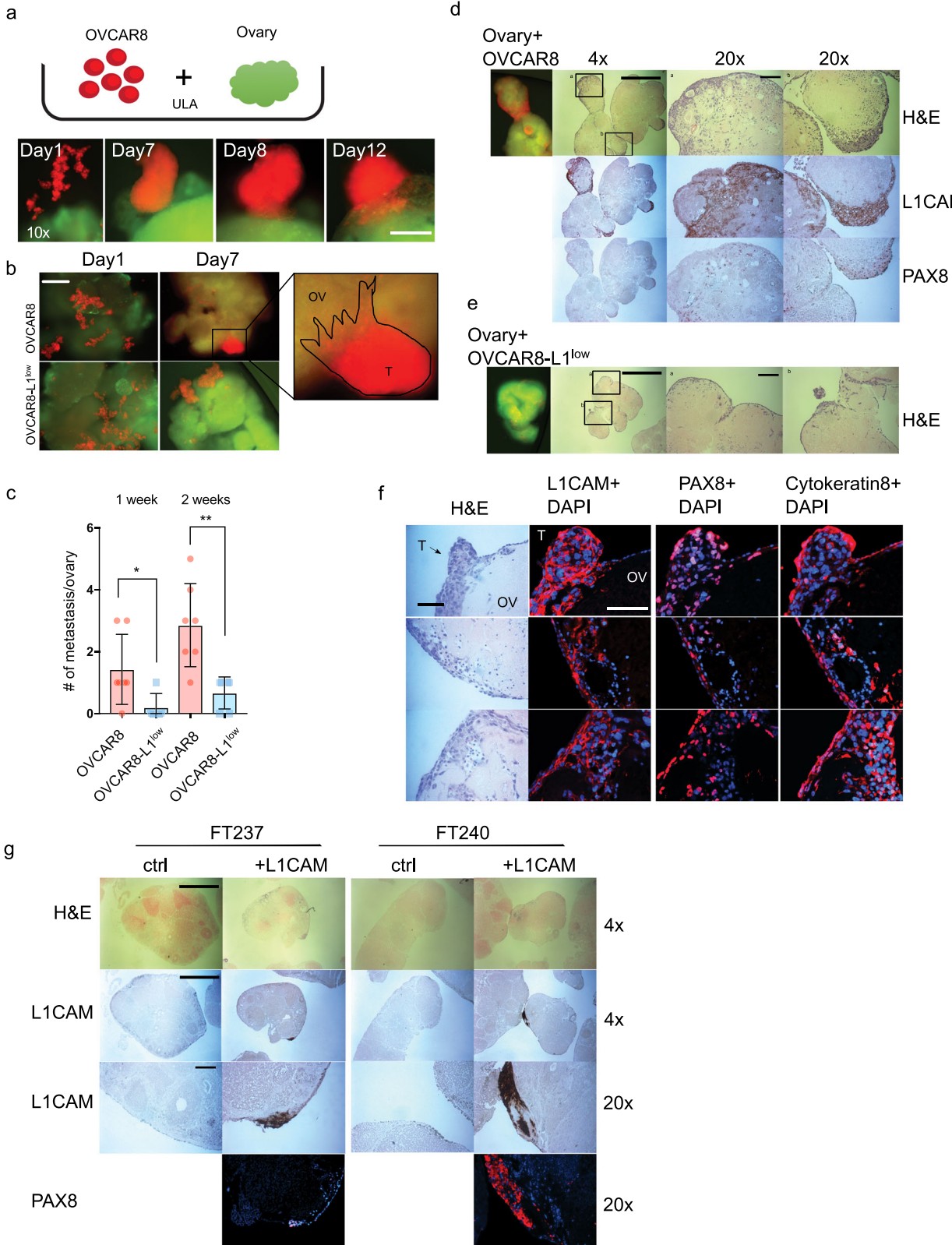

buffer (Crystal Violet 0.05% w/v, Formaldehyde 1%, 10X PBS (1X), Methanol 1%) for 20 min. Cells were washed 5 times with water and air dried. Colonies were counted and size was calculated with ImageJ.

**Overexpression**. We transduced cells transiently with adenoviral particles (Vectorbuilder) generated from a 2nd generation adenovirus gene expression vector containing either hL1CAM [ORF026248] under a CMV promoter or the control vector 72 h before subsequent experiments were performed.

Cells were lysed in RIPA buffer (25 mM Tris-HCl pH 7–8, 150 mM NaCl, 0.1% SDS, 0.5% sodium deoxycholate, 1% Triton X-100 and protease inhibitors) for 30 min on ice. Protein content was quantified by BCA assay using the Pierce BCA kit protocol (#23228). 30 µgs of cell lysate were separated on a 4–15% gradient SDS-PAGE before being transferred to a PVDF membrane using the TurboBlot system (Bio-Rad). The membrane was blocked with 5% nonfat milk in TBST (Tris-buffered saline, 0.1% Tween 20) for one hour at room temperature. Blots were incubated in primary antibody (diluted in blocking buffer) overnight at 4 °C. After

**Fig. 8 L1CAM is important for ovary invasion. a** Upper row: Experiment schematic of co-culture of cells with an isolated mouse ovary under ULA culturing conditions. Lower row: representative observed invasion sequence over time. Scalebar represents 100 μm. **b** Representative fluorescent image after one week's co-culture of OVCAR8 or OVCAR8-L1$^{low}$ (red) with isolated ovary (green) under ULA culture conditions. Scalebar represents 500 μm. **c** Bar graph representation of various OVCAR8 cells metastases to the ovary (more than 10 cells) after 1 and 2 weeks of co-culture. Three biological experiments were performed with at least 3 ovaries for analysis. **d, e** Fluorescent and IHC images representing co-culture of OVCAR8 or OVCAR8-L1$^{low}$ with ovary, respectively. **f** Fluorescent images of ovaries co-cultured with OVCAR8wt for expression of L1CAM, PAX8 and Cytokeratin 8. Scalebar represents 100 μm. **g** H&E and IHC images representing ovaries co-cultured with FT237 or FT246 transfected with L1CAM or a control vector. **d, e** and **g** Scalebar represents 100 μm for 20x objective and 1000 μm for 4x objective.

washing, membranes were incubated in either anti-mouse or anti-rabbit HRP-conjugated secondary antibody (Cell Signaling, 1:1000) diluted in TBST for 1 h. Proteins were detected using Clarity Chemiluminescent HRP Antibody Detection Reagent (Bio-Rad, #1705061) and visualized with a ChemiDoc imaging system (Bio-Rad).

**Viability assay**. Cell viability was monitored after 72 h, using a CellTiter 96® Non-Radioactive Cell Proliferation Assay (Promega, Mannheim, Germany). Each assay was performed in triplicate and repeated at least 3 times. Data are presented by means ± SD. Statistical and significant differences were determined by ANOVA with post-hoc analysis. Cells were additionally stained with crystal violet to count the remaining attached cells.

**Flow cytometry**. Trypsinized cells were washed with FACS buffer (PBS containing 2% BSA) and incubated for 1 h with mAb L1 -11A against L1CAM or a IgG isotype control antibody. After two times washing, the cells were incubated with a secondary anti mouse antibody that was conjugated with Alexa488. After three washing steps, the stained cells were analyzed with a FACS Canto II, using Flowing software (Cell Imaging and Cytometry core and Biocenter, Finland) and the R software. Positive cells were calculated relative to the analyzed isotype control.

**Short-interfering RNA transfection**. A pool of three short-interfering RNA (siRNA) duplexes of the trilencer-27 siRNA (Origene) was used to downregulate corresponding protein expression (Supplementary data 4). As a negative control, a non-specific scrambled trilencer-27 siRNA was used. Twenty-four hrs. before transfection, $1 \times 10^5$ cells were seeded in six-well plates (except OVCAR3, where $2 \times 10^5$ cells were used). Transfection of siRNA was carried out using Lipofectamine RNAiMax (Invitrogen) together with 10 nM siRNA duplex per manufacturer's instructions. To transfect 10 cm plates we scaled up our protocol by a factor of five. Conversely, to transfect one well of a 96-well plate we divided our protocol by 20.

To stably silence RNF20 in FT190 and FT194, cells were transduced with lentiviral vectors (Mission, Sigma-Aldrich) encoding two separate shRNAs: shRNF20_692 (TRCN00000692) or shRNF20_890 (TRCN0000890), or a nontargeting control shRNA: shNTC (SHC002V). The cells were transduced at an MOI of 40 followed by antibiotic selection with puromycin.

**RNA sample processing and sequencing**. RNA was isolated using the RNeasy Plus Mini Kit (Qiagen) as described in the manufacturers protocol. Only RNA samples with high quality and high purity (OD 260/280 = 1.8–2.0) were used to generate libraries using the Bioo Scientific NEXTflex Rapid Directional RNA-seq Library Prep Kit (Bioo Scientific). The resulting mRNA libraries were then sequenced at the JWCI Sequencing Center on an Illumina HiSeq 2500 in Rapid Mode using 101 bp paired-end reads.

RNA was sequenced from FT190 and FT194 cell lines transduced with two separate shRNAs: shRNF20_692 (TRCN00000692) or shRNF20_890 (TRCN0000890), or a nontargeting control shRNA: shNTC (SHC002V) as previously described[25]. Regulated genes (genes with an adjusted p-Value < 0.01) for each cell line and each shRNA were analyzed with the GOrilla platform (Eden et al., 2009, http://cbl-gorilla.cs.technion.ac.il/). The 7 overlapping pathways out of the 25 most enriched in each condition were further characterized.

**Quantitative RT-PCR**. The RNeasy Micro Kit (QIAGEN) was used to isolate RNA. It was reverse transcribed into cDNA using the qScript cDNA Synthesis Kit (Quanta). The 7900HT Fast Real-Time PCR system (Life Technologies) was used to quantify cDNA levels. Triplicate samples were quantified along with minus-RT and minus-template controls. Amplification was continued for 40 cycles as follows: 94 °C for 10 s, 55 °C for 15 s, and 65 °C for 30 s. Relative gene expression was determined by normalizing to the endogenous control, and the resulting value was divided by the central median value of the particular gene across all analyzed samples in the experiment. The following PCR probes were used in these studies: FN1-FOR: GAA CTA TGA TGC CGA CCA GAA, FN1-REV: GGT TGT GCA GAT TTC CTC GT, ITGA5-FOR: CCC ATT GAA TTT GAC AGC AA, ITGA5-REV: TGC AAG GAC TTG TAC TCC ACA

**Immunofluorescence**. For immunofluorescent analysis, cells were grown overnight in a 96-well Cell Imaging Plates (Eppendorf). Cells were then fixed in 4% (v/v) paraformaldehyde in PBS for 20 min at room temperature. Cells were blocked with super-block buffer (Thermo Scientific) and incubated with primary antibody overnight at 4 °C (Supplementary data 5). The secondary antibody was incubated for 1 h. at room temperature. Detection was performed using secondary antibodies conjugated to Cy3 and Alexa 488 Fluor Dyes (Molecular Probes). Cells were then stained with DAPI and after an additional wash Flouromount-G (Sigma-Aldrich) was added. Cells were analyzed by microscopy using a Nikon E400 microscope.

**Immunohistology**. Immunohistochemical staining was performed using Envision Plus/Horseradish Peroxidase system (DAKO, Carpinteria, CA, USA). Formalin-fixed paraffin-embedded tissue sections were de-waxed, rehydrated, and incubated in hydrogen peroxide solution for 30 min to block endogenous peroxidase activity. Antigen retrieval was carried out at 100 °C treatment in citrate buffer (pH 6.0) for 20 min. Sections were incubated with primary antibody overnight at 4 °C (Supplementary data 5). The secondary antibody was applied for 30 min, followed by 3,3'-Diaminobenzidine (DAB) for 5 min.

**Statistics and reproducibility**. All correlation values were calculated with the Pearson correlation coefficient. P-values were calculated with an unpaired two-sided $t$-test. $^*p < 0.05$, $^{**}p < 0.01$, $^{***}p < 0.001$, $^{****}p < 0.0001$.

To calculate the significance of multiple comparison we used the ANOVA test and the Dunnett multiple comparison test to calculate pairwise significance to one control.

*Study approval*. All human and animal protocols were approved by institutional review boards at the University of Pennsylvania.

**Reporting summary**. Further information on research design is available in the Nature Portfolio Reporting Summary linked to this article.

## Data availability

The RNA-seq data presented in this study were submitted to the Gene-Expression Omnibus and can be accessed by the accession number GSE122238 as described in Hooda et al.[25]

The RPPA data are available through the figshare website and the link https://doi.org/10.6084/m9.figshare.12264404.v1. Uncropped and unedited images are available in Supplementary figure 8. Source data for main and supplementary figures are provided as Supplementary data 6.

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

## Acknowledgements

We thank members of the Drapkin lab for fruitful discussions and comments. We thank Paul Whittaker for critical review of the manuscript and Teri Ord for assistance with the isolation of the mouse ovaries. This work was supported by the German Research Foundation DFG (K.D.), a Ruth L. Kirschstein NIH Postdoctoral Individual National Research Service award F32CA221093 (P.T.K.), the Stanley Parker Foundation (S.S.), the Foundation for the Improvement of Research in Gynecology and Obstetrics in Lausanne University Hospital (S.S.), the Dr. Miriam and Sheldon G. Adelson Medical Research Foundation (R.D., G.B.M.), the Honorable Tina Brozman Foundation for Ovarian

Cancer Research (R.D.), The Basser Center for BRCA (R.D.), the Claneil Foundation (R.D.), Ovarian Cancer Research Alliance (J.H., G.B.M), and NIH ovarian cancer SPORE grants P50CA217685 (G.B.M.) and P50CA228991 (R.D.).

## Author contributions

Author contributions: K.D., and R.D designed research; K.D., R.S., H.D.R., S.S., M.P.I., M.F., J.H., L.E.S, K.M.D, P.T.K. and Y.F. performed research; G.B.M. and P.A. contributed new reagents/analytic tools; M.F., P.A., K.D., and R.D. analyzed data; K.D., H.D.R., and R.D. wrote the paper, and all authors reviewed and edited the manuscript.

## Competing interests

R.D. serves as consultant/scientific advisory board member of Repare Therapeutics, Mersana Therapeutics, and VOC Health. GBM receives support or acts as a consultant for: AstraZeneca, ImmunoMET, Ionis, Nanostring, PDX Pharmaceuticals, Signalchem Lifesciences, Symphogen, and Tarveda. No other authors report potential conflicts of interest.
