## [Peer Review File · Communications Biology]

Reviewers' comments:

Reviewer #1 (Remarks to the Author):

L1CAM is required for early dissemination of fallopian tube carcinoma precursors to the Ovary

Doberstein et al.

Manuscript Number: 20-1073-T?

In this study Doberstein and colleagues demonstrates the role of L1CAM in the early dissemination and invasion of the FTSEC to the ovarian surface through promoting survival and extracellular matrix mechanisms. Given that the genetic events that mediate FT lesion metastasis to the ovary is not well understood, the currently study has the potential to make a significant contribution to the understanding of this essential stage in ovarian cancer progression. This is an intriguing study based on strong overall data that incorporates multiple experimental approaches and strategies to address their hypotheses. While the overall paper is strong, I have several concerns which I have outlined below:

Concerns:

1. In Figure 2, the authors have identified differentially expressed genes/pathways following shRNA mediated silencing of RNF20 in FT190 and FT194 cell lines. It was unclear how these studies were completed as no experimental information was provided. This includes the number of replicate samples for each cell line/shRNA that were sequenced, how the sequencing data were processed and analyzed, and/or the statistic approaches used to identify differentially expressed genes or pathways. In addition, the specific genes and pathways identified by these analyses, including the 25 most differentially expressed pathways highlighted in Fig 2a, should be reported. As a result, it was difficult to interpret these data. It was also unclear if the altered pathways found in each experimental condition were due to changes in the same subset of genes in each cell line/shRNA or whether this was a unique set of genes in each. It may be interesting to the reader for the investigators to visualize subsets of commonly altered genes that correspond to specific altered signaling pathways/ cellular processes. For the gene set analysis (Fig 2b), the authors may consider presenting the $-\log_{10}$ transformed p-value as this may provide a more easily readable graph. Finally, the authors should make the GEO accession number available for these RNAseq data as this will be a useful dataset for other investigators.

2. The results presented in Figure 4a-f are not convincing. Given that the tumor cell lines (4A), patient ascites samples (4B) and FTSEC cell lines from health donors (4C) are each on a different blot, it is difficult to compare between cell types. In addition, there does not appear to be a strong correlation between L1CAM expression by western blot (4A-C) and IF (4D-F). As a result of this lack of correlation, it is difficult to assess whether there is truly increased expression in tumor cells vs. FT cells. Although this concern is somewhat mitigated by other data, higher magnification of IF images and/or quantification of WB could address these concerns. In addition, the version of the figures that I have, it is not clear that L1CAM is specifically staining at the membrane of the cell lines as suggested by the authors.

3. For experiments depicting altered migration correlating to altered L1CAM expression (Fig 4G-H), there appears to be a strong negative effect on migration following shRNA-mediated silencing of L1CAM. However, the overexpression data is not especially convincing with less than a 2-fold change in migration. The authors should present western blot data showing the fold increase expression of L1CAM in these cells; relative changes in L1CAM were shown for knock-down studies. Consistent with these concern, can the authors demonstrate that this level of overexpression affects expression of common migration marker proteins in the general cell population over the same time course as

depicted in the transwell migration assays? This is shown later (Fig 7) but it is not clear that the same level of L1CAM overexpression was achieved in each set of experiments.

4. In Figure 7G, the control FT237 cell line seems to have low endogenous expression of Integrin $\alpha 5$. However the data presented in Fig 7H for control FT237 cells indicate that integrin $\alpha 5$ siRNA-mediated silencing causes a small, but statistically significant reduction in sphere size. This raises the concern that there may be off-target effects of the siRNA. The authors should consider using multiple siRNA or shRNA to rule out this possibility.

Minor Concerns:

1. In Figure 4G-H, it is unclear what is being represented on the y-axis. Why is the scale different between each cell line in Fig 4G, while being consistent for Fig 4H? Could the investigators clarify how the data were normalized?

2. In Figure 6E, there appears to be significantly more, larger colonies in the L1CAM+antiL1 9.3 when compared to the L1CAM only cells. These specific images do not appear to correspond with the quantification presented in the figure.

3. The FN1 blot in Figure 7G for the FT237 cell line is either cropped oddly with the main portion of the band missing or is smeared and incomplete. These data would be more convincing with a better quality image.

Reviewer #2 (Remarks to the Author):

Doberstein and colleagues present an interesting body of work that implicates L1CAM as an important protein in the ability of early lesions in fallopian tubes to seed the ovarian surface, eventually leading to ovarian cancer. The data generated from the models used support this theory that has been in the literature for some time now that ovarian cancer does not, in fact, arise in the ovaries, but in cells in the fallopian tube that then shed onto the ovarian surface. The group led by the senior author has made seminal contributions to this field. The novel aspect to this work is implicating L1CAM and the potential therapeutic options this raises.

Points to address:

(1) The authors used shRNA against the E3 ubiquitin ligase RNF20 that writes the histone modification H2Bub1. As has been shown previously in the published literature, this leads to a decrease in H2Bub1. When these knockdown cells were profiled with RNA-seq, it showed upregulation of specific pathways, with one of the factors in common between a subset of these pathways being upregulation of L1CAM. While the correlation is made between down-regulation of RNF20 leading to down-regulation of H2Bub1 and upregulation of L1CAM, the actual mechanism has not been shown that would directly link these events. ChIP-qPCR should be undertaken on coding sequences of L1CAM to investigate what is happening with H2Bub1 at this specific gene loci in response to RNF20 depletion. It might be expected that H2Bub1 will be depleted under these conditions; however, this is counterintuitive to this gene then being highly expressed. This needs to be addressed. Another possibility is that the RNF20-L1CAM relationship relies on an alternative RNF20 substrate, not H2Bub1. Alternative substrates have been published. A third possibility is that the effect of RNF20 on L1CAM is indirect, ie. that its depletion may be causing downregulation of a gene via H2Bub1 loss that may ordinarily negatively regulate L1CAM. All of these possibilities should be discussed, in addition to an experiment performed to directly illuminate the presence/absence of H2Bub1 at the L1CAM locus.

(2) CRISPR-Cas9 gene editing is performed to knockout L1CAM (GeneArt CRISPR system,

ThermoFisher). Can sequences of the resulting clones be included (Supplementary data would be fine).

(3) Why were similar gene editing techniques not used to completely knockout RNF20? This was depleted using shRNA. The published growth of Rnf40^{-/-} MEFs would suggest that a complete RNF20 knockout should be viable?

COMMSBIO-20-1073-T
Response to Referees

We thank the reviewers for their thoughtful and constructive comments. They were invaluable in helping us address deficiencies, add clarity, and improve the manuscript. We have addressed all points raised and they are included in this rebuttal and in the manuscript.

REVIEWER #1:

In this study Doberstein and colleagues demonstrates the role of LICAM in the early dissemination and invasion of the FTSEC to the ovarian surface through promoting survival and extracellular matrix mechanisms. Given that the genetic events that mediate FT lesion metastasis to the ovary is not well understood, the currently study has the potential to make a significant contribution to the understanding of this essential stage in ovarian cancer progression. This is an intriguing study based on strong overall data that incorporates multiple experimental approaches and strategies to address their hypotheses. While the overall paper is strong, I have several concerns which I have outlined below:

Response: We appreciate the reviewer's positive response to our paper.

Concerns:

1. In Figure 2, the authors have identified differentially expressed genes/pathways following shRNA mediated silencing of RNF20 in FT190 and FT194 cell lines. It was unclear how these studies were completed as no experimental information was provided. This includes the number of replicate samples for each cell line/shRNA that were sequenced, how the sequencing data were processed and analyzed, and/or the statistic approaches used to identify differentially expressed genes or pathways.

Response: We thank the reviewer for pointing this out. As mentioned above, we have included additional details that we neglected to incorporate from the Hooda et al. paper. In short, we used two immortalized fallopian tube cell lines (FT190 and FT194) that were transduced with two different shRNAs against RNF20 or a non-target control. The subsequent statistical approaches are now included in the manuscript; regulated genes (genes with an adjusted p-Value <0.01) for each cell line and each shRNA were analyzed with the GOrilla platform (Eden et al., 2009, <http://cbl-gorilla.cs.technion.ac.il/>). The 7 overlapping pathways out of the 25 most enriched in each condition were then displayed. These details are now included in the Methods.

In addition, the specific genes and pathways identified by these analyses, including the 25 most differentially expressed pathways highlighted in Fig 2a, should be reported. As a result, it was difficult to interpret these data. It was also unclear if the altered pathways found in each experimental condition were due to changes in the same subset of genes in each cell line/shRNA or whether this was a unique set of genes in each. It may be interesting to the reader for the investigators to visualize subsets of commonly altered genes that correspond to specific altered signaling pathways/ cellular processes. For the gene set analysis (Fig 2b), the authors may consider presenting the $-\log_{10}$ transformed p-value as this may provide a more easily readable graph.

Finally, the authors should make the GEO accession number available for these RNAseq data as this will be a useful dataset for other investigators.

Response: We agree with the reviewer's suggestion and have made a supplementary table that lists all significantly regulated pathways in each cell line and each shRNA to summarize this information (Supplemental Table 5). In a separate table, we included the differentially expressed genes (genes with an adjusted p-Value <0.01) in each condition and the genes that were included in the cell adhesion, biological adhesion and developmental process pathways in each condition (Supplemental Table 4). As the reviewer suggested, we also included a list of genes that are commonly altered in all conditions. We revised Figure 2b to clarify the scale labeling which depicts the FDR-p values given in log10. The RNA-seq data presented in this study were submitted to the Gene-Expression Omnibus and can be accessed by the accession number GSE122238, as mentioned now in the manuscript.

2. The results presented in Figure 4a-f are not convincing. Given that the tumor cell lines (4A), patient ascites samples (4B) and FTSEC cell lines from healthy donors (4C) are each on a different blot, it is difficult to compare between cell types.

Response: To directly compare the expression between FTSEC cell lines and tumor cell lines, we added a blot to Figure 4 showing a panel (D) that contained both immortalized FT cells and ovarian cancer cell lines. We also included densitometry measurements for each blot to aid comparisons.

In addition, there does not appear to be a strong correlation between LICAM expression by western blot (4A-C) and IF (4D-F). As a result of this lack of correlation, it is difficult to assess whether there is truly increased expression in tumor cells vs. FT cells.

Response: We attribute these differences to the read-outs of each of the methodologies used in each of the respective experiments. Briefly, our western blots represent total LICAM protein whereas the IF represents only LICAM that is accessible on the cell surface. We also observe heterogeneous expression of LICAM expression in ovarian cancer cell lines and FT cell lines. We removed panels E and F to minimize any confusion.

Although this concern is somewhat mitigated by other data, higher magnification of IF images and/or quantification of WB could address these concerns.

Response: We thank the reviewer for this suggestion and have added new IF images for the cancer cell lines and quantified the western blots through densitometry. To better quantify the surface expression of LICAM we have also added the flow cytometry (FACS) data of 3 FT cell lines and 3 cancer cells lines to the supplementary data (Supplemental Figure 1).

In addition, the version of the figures that I have, it is not clear that LICAM is specifically staining at the membrane of the cell lines as suggested by the authors.

Response: The IF was performed without detergent (Triton X100) for permeabilization. Therefore, the antibodies should only bind to surface LICAM. We have clarified this in the manuscript.

3. For experiments depicting altered migration correlating to altered L1CAM expression (Fig 4G-H), there appears to be a strong negative effect on migration following shRNA-mediated silencing of L1CAM. However, the overexpression data is not especially convincing with less than a 2-fold change in migration. The authors should present western blot data showing the fold increase expression of L1CAM in these cells; relative changes in L1CAM were shown for knock-down studies. Consistent with these concerns, can the authors demonstrate that this level of overexpression affects expression of common migration marker proteins in the general cell population over the same time course as depicted in the transwell migration assays? This is shown later (Fig 7) but it is not clear that the same level of L1CAM overexpression was achieved in each set of experiments.

Response: To address this concern, we have added the western blot for the overexpression to the supplementary figures (Supplementary Figure 1). We agree that the effects on migration by overexpression are not very strong, which might be due to the low endogenous expression in some of the cell lines. Since L1CAM is a relatively stable protein, with a low turnaround, and the fact that the migration occurred in only 16h, we would not expect significant changes in L1CAM expression during this time. For instance, in our experience the FT33+Ras cell line does not migrate efficiently. However, in the presence of L1CAM that was enhanced. The other cell lines are capable of transwell migration that is less pronounced, but still significant when L1 is overexpressed.

4. In Figure 7G, the control FT237 cell line seems to have low endogenous expression of Integrin $\alpha 5$. However, the data presented in Fig 7H for control FT237 cells indicate that integrin $\alpha 5$ siRNA-mediated silencing causes a small, but statistically significant reduction in sphere size. This raises the concern that there may be off-target effects of the siRNA. The authors should consider using multiple siRNA or shRNA to rule out this possibility.

Response: The reviewer is correct. Due to high Integrin alpha 5 expression after the overexpression of L1CAM, the low endogenous expression of Integrin alpha 5 cannot be visualized. To address this, we have added a longer exposure of the same western blot (Supplementary Figure 7), showing the endogenous Integrin alpha 5 expression. We also repeated the knockdown of Integrin alpha 5 in FT237 cells to show that its knockdown alone leads to smaller organoids and added this to the supplementary figure section (Supplementary Figure 7).

Minor Concerns:

1. In Figure 4G-H, it is unclear what is being represented on the y-axis. Why is the scale different between each cell line in Fig 4G, while being consistent for Fig 4H? Could the investigators clarify how the data were normalized?

Response: We thank the reviewer for pointing this out. Figure 4H and G are now both normalized to the mean value of the respective control.

2. In Figure 6E, there appears to be significantly more, larger colonies in the L1CAM+antiL1 9.3 when compared to the L1CAM only cells. These specific images do not appear to correspond with the quantification presented in the figure.

Response: We have now changed this figure to include fluorescence pictures that better reflect the measured values.

3. The FN1 blot in Figure 7G for the FT237 cell line is either cropped oddly with the main portion of the band missing or is smeared and incomplete. These data would be more convincing with a better quality image.

Response: Due to the different glycosylated forms of FN1 and its large size of 250kDa, the western blot bands for this protein tend to smear, making it difficult to retrieve a clear band.

REVIEWER #2:

Doberstein and colleagues present an interesting body of work that implicates L1CAM as an important protein in the ability of early lesions in fallopian tubes to seed the ovarian surface, eventually leading to ovarian cancer. The data generated from the models used support this theory that has been in the literature for some time now that ovarian cancer does not, in fact, arise in the ovaries, but in cells in the fallopian tube that then shed onto the ovarian surface. The group led by the senior author has made seminal contributions to this field. The novel aspect to this work is implicating L1CAM and the potential therapeutic options this raises.

Response: We appreciate the reviewer's positive comments.

Points to address:

1. The authors used shRNA against the E3 ubiquitin ligase RNF20 that writes the histone modification H2Bub1. As has been shown previously in the published literature, this leads to a decrease in H2Bub1. When these knockdown cells were profiled with RNA-seq, it showed upregulation of specific pathways, with one of the factors in common between a subset of these pathways being upregulation of L1CAM. While the correlation is made between down-regulation of RNF20 leading to down-regulation of H2Bub1 and upregulation of L1CAM, the actual mechanism has not been shown that would directly link these events. CHIP-qPCR should be undertaken on coding sequences of L1CAM to investigate what is happening with H2Bub1 at this specific gene loci in response to RNF20 depletion.

Response: We appreciate the reviewer's excellent suggestion. While elucidation of the molecular mechanism linking RNF20/H2bUb1 to L1CAM expression was not the goal of this study, we agree that the mechanism by which RNF20/H2Bub1 regulates the L1CAM expression is important and interesting. Therefore, we undertook multiple approaches (as described in response to the Editor and below) to try and define a mechanism. First, we asked whether the L1CAM locus was more accessible (via ATAC-Seq) after RNF20/H2Bub1 KD. Surprisingly, it was not. Second, we asked whether RNF20/H2Bub1 KD impacted DNA methylation around the L1CAM locus in a manner that would influence its expression as described in endometrial and colorectal cancer (Schirmer et al., Kato et al.). We treated cells with 5-AZA (a DNMT inhibitor that leads to demethylation)

and asked whether L1CAM expression was elevated. Surprisingly, 5-AZA treatment led to a decrease in L1CAM expression, the opposite of what we observe with RNF20 silencing. *We suspect that L1CAM may be regulated by a distant enhancer and this is an area of continued research in the lab.*

It might be expected that H2Bub1 will be depleted under these conditions; however, this is counterintuitive to this gene then being highly expressed. This needs to be addressed.

Response: It is important to point out that H2Bub1 status has context dependent effects on transcription. For instance, in basal-like breast cancer cells, silencing RNF20 leads to increased proliferation, migration, and tumorigenicity, through upregulation of inflammatory cytokines (Tarcic et al., Cell Death Differ 2017). This is consistent with a tumor suppressive role of RNF20 and H2Bub1 and our recently published data in ovarian cancer precursor cells (Hooda et al., Can Res 2019). Conversely, in luminal breast cancer cells, silencing RNF20 leads to reduced proliferation, migration, and tumorigenicity, in part through compromised estrogen receptor transcriptional activity. Hence, RNF20 and H2Bub1 have contrasting roles in distinct tumor types, through differential regulation of key transcriptional programs. In our hands, silencing RNF20 and H2Bub1 leads to increased migration and clonogenic growth that is associated with a general increase in chromatin accessibility throughout the genome and particular upregulation of immune modulators and cell adhesion programs. Therefore, it is not necessarily expected that loss of H2Bub1 at the L1CAM locus would lead to its silencing. In fact, we see the opposite.

With regards to ChIP-Seq for H2Bub1, we tried to identify possible binding sites for H2Bub1 in publicly available datasets, such as GSE55921 (Nagarajan S, Hossan T, Alawi M, Najafova Z et al. Bromodomain protein BRD4 is required for estrogen receptor-dependent enhancer activation and gene transcription. Cell Rep 2014 Jul 24;8(2):460-9. PMID: 25017071) that was performed in the breast cancer cell line MCF7, but the dataset lacked data for chromosome X (L1CAM is encoded on Xq28). Nevertheless, we do believe that this is an important experiment and we are currently planning to do a ChIP-seq experiment to study multiple aspects of H2Bub1 for a future publication.

Another possibility is that the RNF20-L1CAM relationship relies on an alternative RNF20 substrate, not H2Bub1. Alternative substrates have been published. A third possibility is that the effect of RNF20 on L1CAM is indirect, ie. that its depletion may be causing downregulation of a gene via H2Bub1 loss that may ordinarily negatively regulate L1CAM. All of these possibilities should be discussed, in addition to an experiment performed to directly illuminate the presence/absence of H2Bub1 at the L1CAM locus.

Response: We thank the reviewer for these excellent suggestions. In addition to looking at the status of DNA accessibility and methylation at the L1CAM locus (above), we also investigated whether known L1CAM regulators or microRNAs may indirectly be impacting L1CAM expression upon RNF20 silencing. We specifically queried our RNA-Seq, RPPA, and ATAC-Seq datasets to

see whether known LICAM regulators such as REST, SNAI2, CTNNB1, PAX8 and AR (as summarized by Altevogt et al. (PMID: 26111503)) are altered by RNF20 silencing. Unfortunately, we identified no correlation between changes in LICAM expression and changes in REST, CTNNB1, SNAI2, PAX8 or AR in any of the datasets. Finally, we analyzed our RNA-seq dataset to identify microRNAs (miRNAs) that are known to regulate LICAM, but did not identify any potential candidate related to RNF20/H2Bub1 loss. Additionally, there was also no overlap in the miRNAs that correlate with LICAM or RNF20 in the TCGA dataset. Nevertheless, we still think that it is possible that a microRNA might regulate LICAM in our context and this is worth exploring in the future.

Overall, we were not successful in identifying the regulatory mechanism after pursuing multiple different angles. However, the main focus of the manuscript is not changed. It remains fixed on describing the central role of LICAM in the dissemination of early cancer cells from the fallopian tube to the ovary. That notwithstanding, our efforts have motivated a more exhaustive approach to address the connection between RNF20, H2Bub1, and LICAM that, while outside the scope of this study, will hopefully lead to further insights.

(2) CRISPR-Cas9 gene editing is performed to knockout LICAM (GeneArt CRISPR system, ThermoFisher). Can sequences of the resulting clones be included (Supplementary data would be fine).

Response: We thank the reviewer for pointing this out. The sequenced knockout clones are now presented in Supplemental Figure 2C and 2D.

In the process of revising this section, we realized that we did not clearly articulate the approach we took to study the role of LICAM in ovarian cancer cell lines. In fact, we took two approaches. One was the CRISPR-Cas9 gene editing approach and the other was to study the behavior of isogenic cell lines with low LICAM vs high LICAM. This latter approach was motivated by the variegated expression of LICAM that we observed in STIC lesions (Figure 2) and with IF in OVCAR8, CAOV3, and OVSAHO cell lines (Figure 4E). Specifically, we saw that some cells expressed high levels of LICAM while neighboring cells might be relatively negative for LICAM. To explore this further, we sorted OVCAR8 cells for high- and low-LICAM expressing cells, using magnetic beads, and characterized these cells as described in Figure 5. To confirm the phenotypes we describe in OVCAR8-LICAM-low cells, we knockout LICAM in the parental OVCAR8 cells and repeated the studies. We used 3 different synthetic sgRNAs and Cas9 protein (both Synthego) to knockout LICAM. Four clones, showing a loss of LICAM protein expression and clear genomic changes (see sup. Fig. 2 C and D) in the desired genomic position were selected for further studies. Functional analysis of our clones revealed that all 4 clones showed the same reduced ability to form organoids and to form colonies (see sup. 2 Fig. E). We believe that the orthogonal approaches with the isogenic line and the 4 CRISPR-Cas9 clones strengthen our findings regarding the importance of LICAM.

(3) Why were similar gene editing techniques not used to completely knockout RNF20? This was depleted using shRNA. The published growth of Rnf40^{-/-} MEFs would suggest that a complete RNF20 knockout should be viable?

Response: We chose to use shRNA against RNF20 for two reasons: (1) Moshe Oren's group reported that homozygous knockout of Rnf20 in mice is embryonic lethal (Tarcic et al., Cell Reports 2016). Similarly, we attempted to knockout Rnf20 in syngeneic murine fallopian tube cells and only got heterozygous clones (unpublished). Therefore, we used shRNAs that we had validated in our prior publication (Hooda et al., Can Res 2019).

Reviewers' comments:

Reviewer #3 (Remarks to the Author):

Primary metastasis of HSOC from the FT to the ovary is any important step in the spread of HSOC that has received little attention in the scientific literature. This paper investigates that metastatic step using bioinformatic, *in vitro*, and *in vivo* experiments. The authors have thoroughly addressed the previous reviewers' comments. However, there are still clarifications needed in the results.

Major Comments

Some figures are not referenced or unclearly referenced in the text, making it frustrating to interpret (see examples below). The authors should carefully check that every figure panel is correctly and clearly referenced in the manuscript.

L115 The heading "L1CAM in early HGSC precursor lesions" seems inappropriate since the next paragraph is about RNF20 and H2Bub1 regulation of L1CAM. That heading would go better at L133.

L120-131 Why did the authors knock down RNF20 and not knock down H2Bub1 directly?

L121 Says you knocked down RNF20 with shRNA but the text does not reference any figure validating the knockdown. It is shown in Figure 2D, but referencing it in text would help the reader immensely.

L129: "To define the specific pathways impacted by the loss of H2Bub1, we used....." You need to be more precise here. The shRNA targeted RNF20 not H2Bub1. I appreciate that RNF20 regulates H2Bub1, but it undoubtedly regulates other genes too. Similarly, L126 states "...L1CAM was among the most significantly upregulated genes upon H2Bub1 knockdown." But again, you knocked down RNF20 not H2Bub1.

L170 states that L1CAM knockdown reduced migration. However, the migration section in the methods say that the Boyden chamber was coated with fibronectin. Usually, when the chamber is coated with an ECM protein, it is usually referred to as Invasion not Migration. Also, indicating how migration was done (i.e. via Boyden chamber) in the results would help readers, so they do not have to go to the Methods to see what time of migration assay was used.

Many places in the results refer to the figures in way where it is unclear what results being shown. For example, L171 states "when we overexpressed L1CAM in the fallopian tube cells with low or moderate expression of L1CAM, we found a significant increase in cell migration (Figure 4G, Supplemental Figure 1C)." The readability of the manuscript would be greatly improved if these types of statements were rewritten so that it was clear that Supplemental Figure 1C confirmed the successful overexpression of L1CAM by western blot and Figure 4G showed L1CAM overexpression increased migration.

L187 states, "the cells remained negative for L1CAM over multiple passages". However, I see no results to support that claim. Figure 5A only shows 1 point in time and days/passages since the cells were sorted is not specified. Supplemental Figure 2B shows control OVCAR8 cells over 6 days, but there is only one image OVCAR8-L1low cells and it is unclear what day it was taken. Regardless there is no mention of passage.

L187 The L1CAM-low OVCAR8 cells significantly reduced the ability to form compact multicellular structures under ultra-low adhesion and adherent culture conditions (Figure 5B, Supplementary Figure 2A)." I don't see how Supplementary Figure 2A supports that statement. It is difficult to see what is happening in that image in the control line, much less in the OVCAR8-low line.

L188 says "... L1CAM-low OVCAR8 cells significantly reduced the ability to form compacted multicellular clusters under ultra-low adhesion and adherent culture conditions (Figure 5B, Supplement Figure 2A)." However, both figures clearly show cells attached to traditional 2D cell

culture plastic. The figure legend for both figures states the cells are in regular 2D culture. Is this ULA portion of this statement referring to Figure 5C?

L196: How many days were cells grown in ULA before imaging? This seems important since you already that OVCAR8-low cells could form spheroids, it just takes more time.

L268 references Figure 7A. L270 references Figure 7D. I do not see where 7B or 7C is referenced. One or two weeks seems like a long time to culture mouse ovaries ex vivo. Do the authors have any insights into the status of the ovaries after such a long incubation? Were the ovaries still viable? Could invasion of cells/spheres into the ovaries be due to breakdown of the ovary?

L345 The summary of your findings are overstated. For example, "The induction of L1CAM is expression in this setting is triggered by loss of histone H2B monoubiquitylation..." However, in this work you show that loss of RNF20 results in loss of HubB1 and overexpression of L1CAM not that HubB1 regulates L1CAM directly. Similarly, "we show that knockout of L1CAM completely abrogated the ability of cancer cells to form spheres" but Figure 5C shows that OVCAR8-low form spheres, just more slowly than control cells.

For ULA experiments, how were the plates made ULA? No description is given. Assuming they purchased as ULA (as opposed to be coated with something in the lab) a company and catalog number would be helpful.

The Statistics states that P-values were calculated with an unpaired two-sided T-test. However, T-tests are inappropriate for experiments with more than 2 treatments (e.g. Figure 5K Figure 6H)

Minor Comments:

L127: should read "significantly upregulated protein" (not gene)

L153: "primary" seems inappropriate since the cells were immortalized

COMMSBIO-20-1073A

Doberstein et al., *L1CAM is required for early dissemination of fallopian tube carcinoma precursors to the ovary.*

Point-by-point rebuttal

Reviewer #3 (Remarks to the Author):

Primary metastasis of HSOC from the FT to the ovary is any important step in the spread of HSOC that has received little attention in the scientific literature. This paper investigates that metastatic step using bioinformatic, in vitro, and in vivo experiments. The authors have thoroughly addressed the previous reviewers' comments. However, there are still clarifications needed in the results.

Response: We thank the reviewer for acknowledging our prior revision.

Major Comments:

1. Some figures are not referenced or unclearly referenced in the text, making it frustrating to interpret (see examples below). The authors should carefully check that every figure panel is correctly and clearly referenced in the manuscript.

Response: We thank the reviewer for pointing this out. We apologize for any confusion. We went through the manuscript carefully to reference each figure appropriately.

2. L115 The heading "L1CAM in early HGSC precursor lesions" seems inappropriate since the next paragraph is about RNF20 and H2Bub1 regulation of L1CAM. That heading would go better at L133.

Response: We thank the reviewer for bringing this to our attention. The reviewer is correct that this paragraph does not fit in L115 and is better suited at L133. We therefore moved the heading to L133 and changed the heading at line L115 to: "**Loss of RNF20 and H2B monoubiquitylation leads to L1CAM upregulation**".

3. L120-131 Why did the authors knock down RNF20 and not knock down H2Bub1 directly? L121 Says you knocked down RNF20 with shRNA but the text does not reference any figure validating the knockdown. It is shown in Figure 2D, but referencing it in text would help the reader immensely.

Response: We thank the reviewer for this comment. It serves as an opportunity for us to more clearly articulate that H2Bub1 is a post-translational modification of histone H2B. Unlike a gene, which can be deleted (CRISPR) or knocked down (snRNA), there is no way to knockdown a specific post-translational modification. However, multiple studies have shown that the heterodimeric RING finger E3 ligase, composed of RNF20 and RNF40, is the dominant ubiquitin ligase responsible for the monoubiquitination of histone H2B. The standard in the field has been

to knockdown RNF20, the catalytic subunit, to study H2Bub1 dynamics and biology. That being said, we agree with the reviewer that the RNF20-RNF40 E3 ligase likely has other targets, though few have been reported in the literature.

We modified the figure and the description in the text to clearly call-out the data in order, as follows:

“Biochemical validation using Western blot analysis showed that depletion of RNF20 resulted in a pronounced reduction in H2Bub1 (Figure 2A). RNAseq enrichment analysis revealed that the knockdown of RNF20 and H2Bub1 resulted in the up-regulation of seven common cellular pathways (Figure 2B, Supplemental Tables 1-2). The seven pathways were enriched in processes related to cell motility, extracellular matrix, and cell adhesion (Figure 2C, Supplemental Table 2). Common to the three pathways related to cell adhesion was L1CAM. In fact, reverse phase protein array (RPPA) analysis revealed that L1CAM was among the most significantly upregulated proteins upon RNF20 and H2Bub1 knockdown (Figure 2D). Western blot analysis showed that depletion of RNF20 resulted in a marked increase in L1CAM in both cell lines (Figure 2E).”

4. L129: “To define the specific pathways impacted by the loss of H2Bub1, we used.....”
You need to be more precise here. The shRNA targeted RNF20 not H2Bub1. I appreciate that RNF20 regulates H2Bub1, but it undoubtedly regulates other genes too. Similarly, L126 states “...L1CAM was among the most significantly upregulated genes upon H2Bub1 knockdown.” But again, you knocked down RNF20 not H2Bub1.

Response: The reviewer is correct. Our comments above explain the rationale for knocking down RNF20. We have changed the sentence to *“In fact, reverse phase protein array (RPPA) analysis revealed that L1CAM was among the most significantly upregulated genes upon RNF20 and H2Bub1 knockdown”*. We also describe in the discussion that other regulations by RNF20 might be responsible for L1CAM expression changes.

5. L170 states that L1CAM knockdown reduced migration. However, the migration section in the methods say that the Boyden chamber was coated with fibronectin. Usually, when the chamber is coated with an ECM protein, it is usually referred to as Invasion not Migration. Also, indicating how migration was done (i.e. via Boyden chamber) in the results would help readers, so they do not have to go to the Methods to see what time of migration assay was used.

Response: We thank the reviewer for this comment. He/she is correct that if an ECM protein is added to the inside of a Boyden chamber, it would be measuring invasion. However, we coated the outside-bottom side of Boyden chamber, that is not in contact with the cells, with fibronectin. The polycarbonate pores were not coated on the inside. Therefore, the cells could directly enter the pores without moving through a matrix. The fibronectin on the outside has the function to keep the cells attached to polycarbonate after migrating through the pore. This makes quantification easier. For an invasion experiment, we would have coated the inner side of the Boyden chamber. In the method section, we have clarified this point by stating “The bottom side

(outside) of the polycarbonate membrane (with a pore size of 5 μm) was pre-coated with fibronectin”

6. Many places in the results refer to the figures in way where it is unclear what results being shown. For example, L171 states “when we overexpressed L1CAM in the fallopian tube cells with low or moderate expression of L1CAM, we found a significant increase in cell migration (Figure 4G, Supplemental Figure 1C).” The readability of the manuscript would be greatly improved if these types of statements were rewritten so that it was clear that Supplemental Figure 1C confirmed the successful overexpression of L1CAM by western blot and Figure 4G showed L1CAM overexpression increased migration.

Response: We thank the reviewer for pointing this out. We have now called out each component separately. We also went through the manuscript to more accurately call-out figure panels.

7. L187 states, “the cells remained negative for L1CAM over multiple passages”. However, I see no results to support that claim. Figure 5A only shows 1 point in time and days/passages since the cells were sorted is not specified. Supplemental Figure 2B shows control OVCAR8 cells over 6 days, but there is only one image OVCAR8-L1low cells and it is unclear what day it was taken. Regardless there is no mention of passage.

Response: We thank the reviewer for mentioning this. Figure 5A represents the expression of L1CAM after 10 passages. We have now included this information in the text. For Supplemental Figure 2B we included only one representative picture because there were no visual morphological differences after the cells reached a confluent monolayer. Moreover, since the OVCAR8-L1^{low} cells show no L1CAM staining, we only see DAPI fluorescence. The image shown in Supplemental Figure 2B was taken on day 6. This is clarified in the figure legend.

8. L187 “The L1CAM-low OVCAR8 cells significantly reduced the ability to form compact multicellular structures under ultra-low adhesion and adherent culture conditions (Figure 5B, Supplementary Figure 2A).” I don’t see how Supplementary Figure 2A supports that statement. It is difficult to see what is happening in that image in the control line, much less in the OVCAR8-low line.

Response: We have corrected the text to properly call-out the figure panels: “The L1CAM-low OVCAR8 cells (termed as OVCAR8-L1^{low}) significantly reduced the ability to form compacted multicellular clusters under adherent culture conditions (Figure 5B, Supplementary Figure 2A) and under ultra-low adhesion conditions (ULA) (Figure 5C).”

9. L188 says “.... L1CAM-low OVCAR8 cells significantly reduced the ability to form compacted multicellular clusters under ultra-low adhesion and adherent culture conditions (Figure 5B, Supplement Figure 2A).” However, both figures clearly show cells

attached to traditional 2D cell culture plastic. The figure legend for both figures states the cells are in regular 2D culture. Is this ULA portion of this statement referring to Figure 5C?

Response: Yes, the reviewer correctly noted that the statement about ULA is referring panel 5C. We have made this clearer in the text.

10. L196: How many days were cells grown in ULA before imaging? This seems important since you already that OVCAR8-low cells could form spheroids, it just takes more time.

Response: We thank the reviewer for pointing this out. We have modified the figure legend (Supp Fig 2) as follows: "(E) upper lane: bright field (BF) and lower lane: RFP signal of OVCAR8 wild type and respective L1CAM knockout clones grown under 3D and serum free conditions. Images captured after 48h."

11. L268 references Figure 7A. L270 references Figure 7D. I do not see where 7B or 7C is referenced.

Response: For clarification, the preceding paragraph describes results in Figure 7A-C. In L268 we refer back to some of those results (panel A and D).

12. One or two weeks seems like a long time to culture mouse ovaries ex vivo. Do the authors have any insights into the status of the ovaries after such a long incubation? Were the ovaries still viable? Could invasion of cells/spheres into the ovaries be due to breakdown of the ovary?

Response: We thank the reviewer for this comment. We shared the same concerns as the reviewer when performing the ex-vivo experiments for the first time. Nevertheless, we did not observe morphological changes by eye or in the microscope when cultivating the ovaries for up to 2 weeks in media. Additionally, when analyzing IHC from the cultured ovaries (Figure 8D, 8F and 8G) we did not observe indication of necrosis or other changes. Most importantly for our experiments was the intactness of the ovarian surface epithelium (Figure 8F).

13. L345 The summary of your findings are overstated. For example, "The induction of L1CAM is expression in this setting is triggered by loss of histone H2B monoubiquitylation..." However, in this work you show that loss of RNF20 results in loss of HubB1 and overexpression of L1CAM not that HubB1 regulates L1CAM directly. Similarly, "we show that knockout of L1CAM completely abrogated the ability of cancer cells to form spheres" but Figure 5C shows that OVCAR8-low form spheres, just more slowly than control cells.

Response: We agree with the reviewer and have modified the language in the discussion accordingly: "*The induction of L1CAM expression in this setting is triggered by loss of RNF20*

and H2Bub1 in early HGSC FT precursors. Second, we show that knockout of L1CAM partially abrogates the ability of cancer cells to form spheres.”

14. For ULA experiments, how were the plates made ULA? No description is given. Assuming they purchased as ULA (as opposed to be coated with something in the lab) a company and catalog number would be helpful.

Response: The plates were purchased from Corning. We have added the relevant information to the Methods section of the manuscript.

15. The Statistics states that P-values were calculated with an unpaired two-sided T-test. However, T-tests are inappropriate for experiments with more than 2 treatments (e.g. Figure 5K Figure 6H)

Response: We thank the reviewer for bringing this to our attention. The reviewer is right and we changed the statistical analysis to perform an ANOVA test. Additionally, for pairwise comparison we calculated the adjusted p-value using the Dunnett's multiple comparison test that compares each of a number of treatments with a single control. We added this information to the Method section and modified Figures 5-7 to reflect this change in statistical analysis.

(Dunnett C. W. (1955). "A multiple comparison procedure for comparing several treatments with a control". Journal of the American Statistical Association.)

Minor Comments:

L127: should read “significantly upregulated protein” (not gene)

Response: We thank the reviewer for mentioning this. The correct has been made.

L153: “primary” seems inappropriate since the cells were immortalized

Response: L153 refers to primary cancer cells derived from ascites fluid of ovarian cancer patients and not the immortalized FT cells.

Reviewer #4's comments:

The paper was significantly improved by the edits from the previous review. The findings are of interest and help to refine pathways that may impact fallopian tube tumor metastasis to the ovary. The only remaining issue is to update parts of the discussion. Several newer papers have been published on factors that drive FTE-ovarian colonization in the last 2 years that seem to be neglected in this version. A host of ovarian factors and several papers have shown that AKT pathway activation primarily from loss of PTEN drives spheroid formation, ovarian adhesion, migration, and invasion. Further, there are several new transgenic mouse models that OVX was shown to impact tumor spread. It would seem highly relevant to integrate this data that LCAM1 can drive AKT activation in the context of the other supporting literature.

Response: We agree with the reviewer with the notion that some papers from the last two years need to be referenced in the discussion.

We therefore added the publication from Joanna Burdette's lab "**Loss of PTEN in Fallopian Tube Epithelium Results in Multicellular Tumor Spheroid Formation and Metastasis to the Ovary**" and from Kim et al. "**In vivo modeling of metastatic human high-grade serous ovarian cancer in mice**" to the manuscript.

We added the following text to the discussion:

"In support with our data, the laboratory of Joanna Burdette has shown in a similar ex-vivo model that the loss of PTEN allows the growth of multicellular tumor spheroids under ultra-low adhesion conditions. Importantly, the loss of PTEN, similarly to the overexpression of L1CAM leads to an activation of the AKT pathway, indicating that the same pathway activation is responsible for an increase in multicellular tumor spheroid formation and metastasis. HGSC transgenic mouse models that include a PTEN loss have been described in multiple studies including a recent study that explores the development and progression of metastatic ovarian cancer. Future studies in a similar transgenic model using overexpression of L1CAM, instead of PTEN loss, would further support the role of L1CAM in ovarian cancer dissemination."

REVIEWERS' COMMENTS:

Reviewer #3 (Remarks to the Author):

The reviewers have adequately addressed my comments.

Reviewer #4 (Remarks to the Author):

The paper is now acceptable for publication.